# Wasserstein Convergence of Score-based Generative Models under Semiconvexity and Discontinuous Gradients

**Stefano Bruno**                                                    *sbruno@ed.ac.uk*
*School of Mathematics*
*University of Edinburgh, United Kingdom*
*Department of Industrial Engineering*
*Ulsan National Institute of Science and Technology, Republic of Korea*

**Sotirios Sabanis**                                                *s.sabanis@ed.ac.uk*
*School of Mathematics*
*University of Edinburgh, United Kingdom*
*National Technical University of Athens, Greece*
*Archimedes, Athena Research Center, Greece*

**Reviewed on OpenReview:** *https://openreview.net/forum?id=vS9iVRB7XF*

## Abstract

Score-based Generative Models (SGMs) approximate a data distribution by perturbing it with Gaussian noise and subsequently denoising it via a learned reverse diffusion process. These models excel at modeling complex data distributions and generating diverse samples, achieving state-of-the-art performance across domains such as computer vision, audio generation, reinforcement learning, and computational biology. Despite their empirical success, existing Wasserstein-2 convergence analysis typically assume strong regularity conditions–such as smoothness or strict log-concavity of the data distribution–that are rarely satisfied in practice. In this work, we establish the first non-asymptotic Wasserstein-2 convergence guarantees for SGMs targeting semiconvex distributions with potentially discontinuous gradients. Our upper bounds are explicit and sharp in key parameters, achieving optimal dependence of $O(\sqrt{d})$ on the data dimension $d$ and convergence rate of order one. The framework accommodates a wide class of practically relevant distributions, including symmetric modified half-normal distributions, Gaussian mixtures, double-well potentials, and elastic net potentials. By leveraging semiconvexity without requiring smoothness assumptions on the potential such as differentiability, our results substantially broaden the theoretical foundations of SGMs, bridging the gap between empirical success and rigorous guarantees in non-smooth, complex data regimes.

## 1 Introduction

Score-based Generative Models (SGMs), also known as diffusion-based generative models (Song & Ermon, 2019; Song et al., 2021; Sohl-Dickstein et al., 2015; Ho et al., 2020), have rapidly emerged over the past few years as a popular approach in modern generative modelling due to their remarkable capabilities in generating complex data, surpassing previous state-of-the-art models, such as Generative Adversarial Networks (GANs) (Goodfellow et al., 2014) and Variational AutoEncoders (VAEs) (Kingma & Welling, 2014). These models are now widely adopted in computer vision and audio generation tasks (Kong et al., 2020; Chen et al., 2021; Mittal et al., 2021; Avrahami et al., 2021; Kim et al., 2021; Bansal et al., 2023; Saharia et al., 2022; Po et al., 2023; Zhang et al., 2023), text generation (Li et al., 2022; Yu et al., 2022; Lovelace et al., 2023), sequential data modeling (Alcaraz & Strodthoff, 2023; Tashiro et al., 2021; Tevet et al., 2023), reinforcement learning and control (Pearce et al., 2023; Chi et al., 2023; Hansen-Estruch et al., 2023; Reuss et al., 2023; Zhu et al., 2023; Ding & Jin, 2024), as well as life-science (Chung & Ye, 2021; Jing et al., 2022; Watson et al., 2023;

Song et al., 2022; Weiss et al., 2023). We refer the reader to the survey papers Yang et al. (2023); Chen et al. (2024) for a more comprehensive exposition of their applications.

The primary goal of SGMs is to generate synthetic data that closely match a target data distribution $\pi_\mathsf{D}$, given a sample set. In particular, these models generate approximate data samples from high-dimensional data distributions by combining two diffusion processes, a forward and a backward process in time. The forward process is used to iteratively and smoothly transform samples from the unknown data distribution into (Gaussian) noise, while the associated backward process reverses the noising procedure to generate new samples from the starting unknown data distribution. A key role in these models is played by the score function, i.e. the gradient of the log-density of the solution of the forward process, which appears in the drift of the stochastic differential equation (SDE) associated with the backward process. Since this quantity depends on the unknown data distribution, an estimator of the score must be constructed during the noising step using score-matching techniques (Hyvärinen, 2005; Vincent, 2011).

The widespread applicability and success of SGMs have been accompanied by a growing interest in the theoretical understandings of these models, particularly in the convergence analysis under different metrics such as Total Variation (TV) distance, Kullback Leibler (KL) divergence, Wasserstein distance, e.g., Block et al. (2020); De Bortoli et al. (2021); Bortoli (2022); Lee et al. (2022); Yang & Wibisono (2022); Kwon et al. (2022); Liu et al. (2022); Oko et al. (2023); Lee et al. (2023); Chen et al. (2023a;b); Li et al. (2024); Pedrotti et al. (2024); Conforti et al. (2025); Benton et al. (2024); Strasman et al. (2025); Bruno et al. (2025); Tang & Zhao (2024); Mimikos-Stamatopoulos et al. (2024); Wang & Wang (2024); Silveri & Ocello (2025); Yu & Yu (2025). In this work, we provide a non-asymptotic convergence analysis in Wasserstein distance of order two, as this metric is often considered more practical and informative for estimation tasks (see e.g., equation 4), and is closely connected to the popular Fréchet Inception Distance (FID) used to assess the quality of images in generative modeling (see, e.g., Section 4). A significant limitation of prior analysis in Wasserstein-2, e.g., Strasman et al. (2025); Gao et al. (2025); Bruno et al. (2025); Tang & Zhao (2024); Wang & Wang (2024); Yu & Yu (2025), is their reliance on strong regularity conditions–such as smoothness or strict log-concavity – of the data distribution and its potential. These assumptions facilitate mathematical tractability but limit the applicability of theoretical results to more general settings, especially when the data distribution is only semiconvex and the potential's gradient may be discontinuous. The only exception outside the strict log-concavity regime is the recent contribution in Silveri & Ocello (2025), where the authors assumes that the data distribution is weakly convex. However, their analysis still requires the potential to be twice continuously differentiable (see, e.g., Silveri & Ocello (2025, Proofs of Propositions B.1 and B.2)), and the stepsize of their generative algorithm must be bounded by a quantity inversely proportional to the one-sided Lipschitz constant of the potential (see Silveri & Ocello (2025, equation (30))). Still, such conditions on $\pi_\mathsf{D}$ in existing Wasserstein-2 convergence analysis do not fully reflect the complexity of real-world data, which often exhibit non-smooth or non-log-concave distributions. Therefore, the aim of this work is to address the following fundamental question:

> *Can Score-based Generative Models be guaranteed to converge in Wasserstein-2 distance when the data distribution is only semiconvex and the potential admits discontinuous gradients?*

We provide a positive answer to this question by combining recent findings in non-smooth, non-log-concave sampling, with standard stochastic analysis tools, thereby presenting the first contributions in the Score-based generative modeling literature for non-smooth potentials. We establish explicit, non-asymptotic Wasserstein-2 convergence bounds for SGMs under semiconvexity assumptions on the data distribution, accommodating potentials with discontinuous gradients. This framework covers a variety of practically relevant distributions arising in Bayesian statistical methods, including symmetric modified half-normal distributions, Gaussian mixtures, double-well potentials, and elastic net potentials, all of which satisfy our relaxed assumptions.

In addition, our estimates are explicit and exhibit the best known optimal dependencies in terms of data dimension, i.e., $O(\sqrt{d})$ in Theorem 19, and rate of convergence, i.e., $O(\gamma)$ in Theorem 21. In contrast to prior works under the same metric Silveri & Ocello (2025); Gao et al. (2025); Strasman et al. (2025); Tang & Zhao (2024), our estimates in Theorem 19 and Theorem 21 are derived without imposing any restrictions

on the stepsize of the generative algorithm[1], making them more suitable for practical implementation. By circumventing the need for strict regularity conditions on the score function and allowing discontinuities in the gradients of the potentials, our work significantly expands the theoretical foundation of SGMs. This advancement not only bridges the gap between empirical success and theoretical guarantees but also opens new avenues for the application of diffusion models to data distributions with non-smooth potentials.

One source of error in the construction of the generative algorithm arises from replacing the initial condition of the backward process with the invariant measure of the forward process. To ensure this error remains small, the drift terms of both SDEs must satisfy, for instance, a monotonicity property with a time-dependent bound that meets an appropriate integrability condition (see, e.g., equation 19 and equation 23 below). To address this, we identify a time horizon for the generative algorithm that ensures the paths of the two backward processes become contractive. Notably, the integrability condition on the monotonicity bound depends only on the known constants in Assumption 2, making it significantly easier to verify in practice compared to the analogous condition in Silveri & Ocello (2025, Appendix C), which relies on weak convexity constants that are often difficult to estimate.

In conclusion, we present the first explicit, dimension- and parameter-dependent $W_2$-convergence guarantees for Score-based Generative models operating on data distributions having potentials with discontinuous gradients. Our results mark an important step forward in the rigorous analysis of SGMs, providing both theoretical insights and practical tools for advancing generative modeling in challenging, non-smooth regimes.

*Notation.* Let $(\Omega, \mathcal{F}, \mathbb{P})$ be a fixed probability space. We denote by $\mathbb{E}[X]$ the expectation of a random variable $X$. For $1 \leq p < \infty$, $L^p$ is used to denote the usual space of $p$-integrable real-valued random variables. The $L^p$-integrability of a random variable $X$ is defined as $\mathbb{E}[|X|^p] < \infty$. Fix an integer $d \geq 1$. For an $\mathbb{R}^d$-valued random variable $X$, its law on $\mathcal{B}(\mathbb{R}^d)$, i.e. the Borel sigma-algebra of $\mathbb{R}^d$ is denoted by $\mathcal{L}(X)$. Let $T > 0$ denote a time horizon. For a positive real number $b$, we denote its integer part by $\lfloor b \rfloor$. The Euclidean scalar product is denoted by $\langle \cdot, \cdot \rangle$, with $|\cdot|$ standing for the corresponding norm (where the dimension of the space may vary depending on the context). Let $f : \mathbb{R}^d \to \mathbb{R}$ be a continuously differentiable function. The gradient of $f$ is denote by $\nabla f$. For any integer $q \geq 1$, let $\mathcal{P}(\mathbb{R}^q)$ be the set of probability measures on $\mathcal{B}(\mathbb{R}^q)$. For $\mu$, $\nu \in \mathcal{P}(\mathbb{R}^d)$, let $\mathcal{C}(\mu, \nu)$ denote the set of probability measures $\zeta$ on $\mathcal{B}(\mathbb{R}^{2d})$ such that its respective marginals are $\mu$ and $\nu$. For any $\mu$ and $\nu \in \mathcal{P}(\mathbb{R}^d)$, the Wasserstein distance of order 2 is defined as

$$W_2(\mu, \nu) = \left( \inf_{\zeta \in \mathcal{C}(\mu, \nu)} \int_{\mathbb{R}^d} \int_{\mathbb{R}^d} |x - y|^2 \, \mathrm{d}\zeta(x, y) \right)^{\frac{1}{2}}.$$

Table 4 (Appendix E) lists the main symbols used throughout this work along with references to where they are defined.

## 2 Technical Background for OU-based SGMs

In this section, we briefly summarize the construction of score-based generative models (SGMs) via diffusion processes, as introduced by Song et al. (2021). The core idea behind SGMs is to employ an ergodic (forward) diffusion process that gradually transforms the unknown data distribution $\pi_\mathsf{D} \in \mathcal{P}(\mathbb{R}^d)$ into a known prior distribution. A backward (in time) process is then learned to transform the prior back to the target distribution $\pi_\mathsf{D}$ by estimating the score function of the forward process. In our analysis, we consider the forward process $(X_t)_{t \in [0,T]}$ to be an Ornstein-Uhlenbeck (OU) process, consistent with the choice in the original paper Song et al. (2021)

$$\mathrm{d}X_t = -X_t \, \mathrm{d}t + \sqrt{2} \, \mathrm{d}B_t, \quad X_0 \sim \pi_\mathsf{D}, \tag{1}$$

where $(B_t)_{t \in [0,T]}$ is an $d$-dimensional Brownian motion and we assume that $\mathbb{E}[|X_0|^2] < \infty$.

---

[1]The results in Silveri & Ocello (2025); Gao et al. (2025); Strasman et al. (2025); Tang & Zhao (2024) require the stepsize to be controlled in terms of the Lipschitz constant or the strong convexity constant of the target data distribution; see, e.g., Table 2 below. The only exceptions are the results in Bruno et al. (2025, Remark 12 and Theorem 10), which, however, requires stronger assumptions on the data distribution than Assumption 2 below.

For target data distributions $\pi_D$ that are absolutely continuous with respect to the Lebesgue measure, and whose densities are continuous and integrable, the backward process $(Y_t)_{t\in[0,T]} = (X_{T-t})_{t\in[0,T]}$ is well defined [2] (Millet et al., 1989; Haussmann & Pardoux, 1986), and is given by

$$\mathrm{d}Y_t = (Y_t + 2\nabla \log p_{T-t}(Y_t))\,\mathrm{d}t + \sqrt{2}\,\mathrm{d}\bar{B}_t, \quad Y_0 \sim \mathcal{L}(X_T), \tag{2}$$

where $\{p_t\}_{t\in[0,T]}$ is the family of densities of $\{\mathcal{L}(X_t)\}_{t\in(0,T]}$ with respect to the Lebesgue measure, $\bar{B}_t$ is an another Brownian motion independent of $B_t$ in 1 defined on $(\Omega, \mathcal{F}, \mathbb{P})$. In practice, however, the initial distribution is taken to be the invariant measure of the forward process, which corresponds to the standard Gaussian distribution. As a result, the backward process in 2 becomes

$$\mathrm{d}\widetilde{Y}_t = (\widetilde{Y}_t + 2\,\nabla \log p_{T-t}(\widetilde{Y}_t))\,\mathrm{d}t + \sqrt{2}\,\mathrm{d}\bar{B}_t, \quad \widetilde{Y}_0 \sim \pi_\infty = \mathcal{N}(0, I_d). \tag{3}$$

Since the target distribution $\pi_D$ is unknown, the score function $\nabla \log p_t$ in 2 cannot be computed exactly. To overcome this limitation, an estimator $s(\cdot, \theta^*, \cdot)$ is *learned* based on a family of functions $s : [0,T] \times \mathbb{R}^M \times \mathbb{R}^d \to \mathbb{R}^d$ parametrized in $\theta$, aiming at approximating the score of the ergodic forward process 4 over a fixed time window $[0,T]$. In practice, $s$ are neural networks and in particular cases, e.g., the motivating example in Bruno et al. (2025, Section 3.1), the functions $s$ can be carefully designed. The optimal value $\theta^*$ of the parameter $\theta$ is determined by optimizing the following score-matching objective

$$\mathbb{R}^d \ni \theta \mapsto \mathbb{E}\left[\int_0^T |\nabla \log p_t(X_t) - s(t, \theta, X_t)|^2\,\mathrm{d}t\right]. \tag{4}$$

An explicit expression of the stochastic gradient of 4 derived via denoising score matching (Vincent, 2011) is provided in Bruno et al. (2025, equation (8), Section 2). Following Bruno et al. (2025, Section 2), we define an auxiliary process $(Y_t^{\mathrm{aux}})_{t\in[0,T]}$ that incorporates the approximating function $s$, which depends on the (random) estimator of $\theta^*$ denoted by $\hat{\theta}$. For $t \in [0,T]$, this process is given by

$$\mathrm{d}Y_t^{\mathrm{aux}} = (Y_t^{\mathrm{aux}} + 2\,s(T-t, \hat{\theta}, Y_t^{\mathrm{aux}}))\,\mathrm{d}t + \sqrt{2}\,\mathrm{d}\bar{B}_t, \quad Y_0^{\mathrm{aux}} \sim \pi_\infty = \mathcal{N}(0, I_d). \tag{5}$$

The auxiliary process 5 serves as a bridge between the backward process 3 and the numerical scheme 7, and it facilitates the analysis of the convergence of the diffusion model (see the upper bounds involving $Y_t^{\mathrm{aux}}$ in the proof of Theorem 19 in Appendix C for further details). We now introduce the numerical scheme. Let the step size $\gamma_j = \gamma \in (0,1)$ for each $j = 0, \ldots, J$, where $J \in \mathbb{N}$. The discrete process $(Y_j^{\mathrm{EM}})_{j\in\{0,\ldots,J+1\}}$ of the Euler–Maruyama approximation of 5 is given, for any $j \in \{0, \ldots, J\}$, as follows

$$Y_{j+1}^{\mathrm{EM}} = Y_j^{\mathrm{EM}} + \gamma(Y_j^{\mathrm{EM}} + 2\,s(T-t_j, \hat{\theta}, Y_j^{\mathrm{EM}})) + \sqrt{2\gamma}\,\bar{Z}_{j+1}, \quad Y_0^{\mathrm{EM}} \sim \pi_\infty = \mathcal{N}(0, I_d), \tag{6}$$

where $\{\bar{Z}_j\}_{j\in\{0,\ldots,J+1\}}$ is a sequence of independent $d$-dimensional Gaussian random variables with zero mean and identity covariance matrix. The continuous-time interpolation of 6, for $t \in [0,T]$, is given by

$$\mathrm{d}\widehat{Y}_t^{\mathrm{EM}} = (\widehat{Y}_{\lfloor t/\gamma \rfloor \gamma}^{\mathrm{EM}} + 2\,s(T - \lfloor t/\gamma \rfloor \gamma, \hat{\theta}, \widehat{Y}_{\lfloor t/\gamma \rfloor \gamma}^{\mathrm{EM}}))\,\mathrm{d}t + \sqrt{2}\,\mathrm{d}\bar{B}_t, \quad \widehat{Y}_0^{\mathrm{EM}} \sim \pi_\infty = \mathcal{N}(0, I_d), \tag{7}$$

where $\mathcal{L}(\widehat{Y}_j^{\mathrm{EM}}) = \mathcal{L}(Y_j^{\mathrm{EM}})$ at grid points for each $j \in \{0, \ldots, J+1\}$.

## 3 Wasserstein Convergence Analysis for SGMs

In this section, we provide the full non-asymptotic estimates in Wasserstein distance of order two between the target data distribution $\pi_D$ and the generative distribution of the diffusion model under the assumptions stated below. As discussed in Bruno et al. (2025, Section 2 and Appendix A), it may be necessary to restrict $t \in [\epsilon, T]$ for $\epsilon \in (0,1)$ in 4 to account for numerical instabilities that can arise during training and sampling near $t = 0$ as also observed in practice in Song et al. (2021, Appendix C), and for the possibility that the integral of the score function in 4 may diverge when $t = 0$. Therefore, we truncate the integration in the backward diffusion at $T - \epsilon$ and consider the process $(Y_t)_{t\in[0,T-\epsilon]}$.

---

[2]The regularity of the Ornstein–Uhlenbeck semigroup for all $t \in (0,T]$ (see Appendix A, and Conforti et al. (2025, Proof of Proposition 3.1)) ensures that the necessary and sufficient conditions for the reversibility of the diffusion process are satisfied; see, e.g., Millet et al. (1989, Theorem 2.2) or (Haussmann & Pardoux, 1986, Theorem 2.1). These conditions on $\pi_D$ are included in Assumption 2.

### 3.1 Assumptions

We begin by stating the main assumptions of our setting. The optimization problem in 4 can be solved using algorithms such as stochastic gradient descent (Jentzen et al., 2021), ADAM (Kingma & Ba, 2015), Stochastic Gradient Langevin Dynamics (Bruno et al., 2025, Section 3.1), and TheoPouLa (Lim & Sabanis, 2024), provided they satisfy the following assumption.

**Assumption 1.** *Let $\theta^*$ be a minimiser[3] of 4 and let $\hat{\theta}$ be the (random) estimator of $\theta^*$ obtained through some approximation procedure such that $\mathbb{E}[|\hat{\theta}|^2] < \infty$. There exists $\widetilde{\varepsilon}_{AL} > 0$ such that*

$$\mathbb{E}[|\hat{\theta} - \theta^*|^2] < \widetilde{\varepsilon}_{AL}.$$

**Remark 1.** *As a consequence of Assumption 1, one obtains $\mathbb{E}[|\hat{\theta}|^2] < 2\widetilde{\varepsilon}_{AL} + 2|\theta^*|^2$.*

In this work, we consider the potentials to be semiconvex functions– a broad generalization of convex functions that includes non-convex functions whose curvature is bounded from below. This class allows for discontinuities in the gradient while retaining key analytical properties of convex functions, such as the existence of well-defined subgradients.

**Definition 2.** A function $U$ is semiconvex if there exists $K \geq 0$ such that $U + \frac{K}{2}|\cdot|^2$ is convex.

Semiconvexity has received significant attention[4] in the machine learning community (Davis et al., 2018; Sun & Yu, 2019; Richards & Rabbat, 2021; Liu et al., 2021; Rafique et al., 2022), optimization (Li et al., 2020; Ma et al., 2020; Li & Xu, 2021; Hu et al., 2025), optimal control (Cannarsa & Sinestrari, 2004), and the study of fully nonlinear partial differential equations (Braga et al., 2019; Payne & Redaelli, 2023). We refer the reader to Duda & Zajıcek (2009); Cattiaux & Guillin (2014) for comprehensive overviews of the mathematical challenges associated with this class. Importantly, semiconvex functions may admit discontinuous gradients which are characterized using the Fréchet subdifferential (Bazaraa et al., 1974; Alberti et al., 1992).

**Definition 3.** For a function $U : \mathbb{R}^d \to \mathbb{R}$, we define the subdifferential $\partial U(x)$ of $U$ at $x \in \mathbb{R}^d$ as

$$\partial U(x) = \left\{ \tilde{p} \in \mathbb{R}^d : \liminf_{z \to x} \frac{U(z) - U(x) - \langle \tilde{p}, z - x \rangle}{|z - x|} \geq 0 \right\}. \tag{8}$$

The set 8 is closed and convex, and may be empty in general. We say that $U$ is Fréchet subdifferentiable at $x$ if $\partial U(x) \neq \emptyset$. Any element $h(x) \in \partial U(x)$ is called a Fréchet subgradient of $U$ at $x \in \mathbb{R}^d$. When $U$ is differentiable at $x$, the subdifferential reduces to $\partial U(x) = \{\nabla U(x)\}$. Crucially, semiconvex functions are Fréchet subdifferentiable at every points in $\mathbb{R}^d$, i.e. $\partial U(x) \neq \emptyset$ for all $x$ (Alberti et al., 1992; Cannarsa & Sinestrari, 2004). In this case, $h(x) + Kx$ corresponds to the classical convex subgradient (Vial, 1983, Proposition 4.6). Moreover, every element of the subdifferential of a semiconvex function satisfies a one-sided Lipschitz condition, ensuring the existence of $h(x) \in \partial U(x)$. The following lemma– adapted from (Alberti et al., 1992, Proposition 2.1) and presented in (Johnston et al., 2025, Lemma 1)–and its subsequent corollary formalize this property.

**Lemma 4.** *(Alberti et al., 1992, Modification of Proposition 2.1) Let $U$ be a semiconvex function. Then, $U$ is locally Lipschitz continuous, the subdifferential set $\partial U$ is non-empty, compact, and $\tilde{p} \in \partial U(x)$, if and only if*

$$U(z) - U(x) - \langle \tilde{p}, z - x \rangle \geq -\frac{K}{2}|z - x|^2,$$

*for all $x, z \in \mathbb{R}^d$.*

**Corollary 5.** *(Johnston et al., 2025, Corollary 1) Let $x, z \in \mathbb{R}^d$, $\tilde{p} \in \partial U(x)$, and $\tilde{q} \in \partial U(z)$. Then,*

$$\langle \tilde{p} - \tilde{q}, x - z \rangle \geq -K|x - z|^2.$$

---

[3]The score-matching optimization problem 4 is not necessarily (strongly) convex.

[4]Some of these references refer to semiconvex functions as weakly convex. We avoid this terminology to prevent confusion with the notion of weak convexity introduced in Definition 10 below.

We state the assumption on the target data distribution $\pi_D$ below. Recall that $h(x) \in \partial U(x)$ is the Fréchet subgradient of $U$ at $x \in \mathbb{R}^d$.

**Assumption 2.** *The data distribution $\pi_D$ has a finite second moment and it is absolutely continuous with respect to the Lebesgue measure with $\pi_D(\mathrm{d}x) = \exp(-U(x))\,\mathrm{d}x$ for some $U : \mathbb{R}^d \to \mathbb{R}$. Moreover,*

*(i) The potential $U$ is continuous and its gradient exists almost everywhere.*

*(ii) The potential $U$ is $K$-semiconvex (on a ball). That is, there exists $K, R \geq 0$, such that for all $x, \bar{x} \in \mathbb{R}^d$,*

$$\langle h(x) - h(\bar{x}), x - \bar{x} \rangle \geq -K|x - \bar{x}|^2, \qquad \text{when} \quad |x - \bar{x}| < R,$$

*(iii) The potential $U$ is $\mu$-strongly convex at infinity[5]. That is, there exists $\mu > 0$ such that for all $x, \bar{x} \in \mathbb{R}^d$,*

$$\langle h(x) - h(\bar{x}), x - \bar{x} \rangle \geq \mu|x - \bar{x}|^2, \qquad \text{when} \quad |x - \bar{x}| \geq R, \tag{9}$$

*where $R$ is the same as in point (ii).*

**Remark 6.** *As a consequence of Proposition 23, due to Conforti et al. (2025, Proposition 3.1), and Assumption 2-(i), we have that, for any $t \in (0, T)$, the map $x \mapsto \nabla p_t(x)$ is continuously differentiable, and for any $x \in \mathbb{R}^d$, the map $t \mapsto p_t(x)$ is continuously differentiable on $(0, T]$. Moreover, Assumption 2 implies that the processes in 2 and 3 have a unique strong solution.*

Next, we consider the following assumption on the approximating function $s$, which is also adopted in Bruno et al. (2025, Assumption 3.a).

**Assumption 3.a.** The function $s : [0, T] \times \mathbb{R}^M \times \mathbb{R}^d \to \mathbb{R}^d$ is continuously differentiable in $x \in \mathbb{R}^d$. Let $D_1 : \mathbb{R}^M \times \mathbb{R}^M \to \mathbb{R}_+$, $D_2 : [0, T] \times [0, T] \to \mathbb{R}_+$ and $D_3 : [0, T] \times [0, T] \to \mathbb{R}_+$ be such that $\int_\epsilon^T \int_\epsilon^T D_2(t, \bar{t})\,\mathrm{d}t\,\mathrm{d}\bar{t} < \infty$ and $\int_\epsilon^T \int_\epsilon^T D_3(t, \bar{t})\,\mathrm{d}t\,\mathrm{d}\bar{t} < \infty$. For $\alpha \in \left[\frac{1}{2}, 1\right]$ and for all $t, \bar{t} \in [0, T]$, $x, \bar{x} \in \mathbb{R}^d$, and $\theta, \bar{\theta} \in \mathbb{R}^M$, we have that

$$|s(t, \theta, x) - s(\bar{t}, \bar{\theta}, \bar{x})| \leq D_1(\theta, \bar{\theta})|t - \bar{t}|^\alpha + D_2(t, \bar{t})|\theta - \bar{\theta}| + D_3(t, \bar{t})|x - \bar{x}|,$$

where $D_1$, $D_2$ and $D_3$ have the following growth in each variable: i.e., there exist $\mathsf{K}_1$, $\mathsf{K}_2$, and $\mathsf{K}_3 > 0$ such that for each $t, \bar{t} \in [0, T]$ and $\theta, \bar{\theta} \in \mathbb{R}^M$,

$$|D_1(\theta, \bar{\theta})| \leq \mathsf{K}_1(1 + |\theta| + |\bar{\theta}|), \qquad |D_2(t, \bar{t})| \leq \mathsf{K}_2(1 + |t|^\alpha + |\bar{t}|^\alpha),$$
$$|D_3(t, \bar{t})| \leq \mathsf{K}_3(1 + |t|^\alpha + |\bar{t}|^\alpha).$$

**Remark 7.** *Assumption 3.a requires that the approximating function $s$ is Lipschitz continuous in both the input variable $x$ and the parameter $\theta$. In time $t$, it allows $s$ to be either Hölder continuous (for $\alpha \in [\frac{1}{2}, 1)$) or Lipschitz continuous (for $\alpha = 1$). This relaxed continuity in $t$ for the drift term of 7 is standard for the Euler–Maruyama schemes for SDEs. Crucially, we show in Theorems 19 and 21 that this weaker condition in $t$ still guarantees convergence of the generative algorithm to $\pi_D$. As noted by Bruno et al. (2025, Remark 6) in the context of neural network-based approximations, Assumption 3.a, when $\alpha = 1$, covers the case where $s$ is implemented as a neural network with a hyperbolic tangent or sigmoid activation function at the final layer. Moreover, Assumption 3.a implies that the process in 5, 6, and 7 have a unique strong solution.*

**Remark 8.** *Let $\mathsf{K}_{Total} := \mathsf{K}_1 + \mathsf{K}_2 + \mathsf{K}_3 + |s(0, 0, 0)| > 0$. Using Assumption 3.a, one obtains*

$$|s(t, \theta, x)| \leq \mathsf{K}_{Total}(1 + |t|^\alpha)(1 + |\theta| + |x|).$$

The proof of Remark 8 can be found, e.g., in Bruno et al. (2025, Appendix D.3). By imposing an additional condition on the gradient of $s$ in Assumption 3.a—as done in Bruno et al. (2025, Assumption 3.b)—, we obtain the optimal convergence rate established in Theorem 21 below.

**Assumption 3.b.** Let $s$ be as in Assumption 3.a and there exists $\mathsf{K}_4 > 0$ such that, for all $x, \bar{x} \in \mathbb{R}^d$ and for any $k = 1, \dots d$,

$$|\nabla_x s^{(k)}(t, \theta, x) - \nabla_{\bar{x}} s^{(k)}(t, \theta, \bar{x})| \leq \mathsf{K}_4(1 + 2|t|^\alpha)|x - \bar{x}|.$$

---

[5]Intuitively, outside a sufficiently large region, $U$ bends upwards at least as much as a quadratic function.

For the following assumption on the score approximation, we let $\hat{\theta}$ be as in Assumption 1 and we let $(Y_t^{\mathrm{aux}})_{t \in [0,T]}$ be the auxiliary process defined in 5.

**Assumption 4.** *There exists $\varepsilon_{SN} > 0$ such that*

$$\mathbb{E} \int_0^{T-\epsilon} |\nabla \log p_{T-r}(Y_r^{aux}) - s(T-r, \hat{\theta}, Y_r^{aux})|^2 \, dr < \varepsilon_{SN}. \tag{10}$$

**Remark 9.** *Assumption 4 is now a standard assumption considered in the literature, see, e.g., Gao et al. (2025); Bruno et al. (2025); Strasman et al. (2025); Silveri & Ocello (2025), and its theoretical and practical soundedness is discussed, e.g., in Bruno et al. (2025, Remark 7, 8, and 9).*

### 3.2 Assumption 2 and Weak Convexity of the Data Distribution

We start by introducing the definition of weak convexity, a concept that has been widely used in conjunction with coupling techniques to analyze the long-time behavior of gradient flows SDEs (Conforti, 2023; 2024; Conforti et al., 2023b;a)[6], and we extend its application here to subgradients of $U$.

**Definition 10.** The potential $U : \mathbb{R}^d \to \mathbb{R}$ is weakly convex if its weak convexity profile $\kappa_U : [0, \infty) \to \mathbb{R}$ defined as

$$\kappa_U(r) = \inf_{x, \bar{x} \in \mathbb{R}^d: \, |x - \bar{x}| = r} \left\{ \frac{\langle h(x) - h(\bar{x}), x - \bar{x} \rangle}{|x - \bar{x}|^2} \right\}, \tag{11}$$

where $h(x) \in \partial U(x)$ is the subgradient of $U$ at $x \in \mathbb{R}^d$, satisfies

$$\kappa_U(r) \geq \beta - r^{-1} f_L(r), \quad \text{for all } r > 0, \tag{12}$$

for some constants $\beta, L > 0$, where the function $f_L : [0, \infty] \to [0, \infty]$ is defined as

$$f_L(r) = 2L^{1/2} \tanh((rL^{1/2})/2). \tag{13}$$

**Remark 11.** *The weak convexity profile $\kappa_U$ in 11 serves as an averaged/integrated convexity lower bound for the potential $U$, evaluated over pairs of points separated by a distance $r > 0$. Unlike the standard convexity condition $\kappa_U \geq 0$, which characterizes convex (or log-concave) potentials, Definition 10 allows $\kappa_U$ to vary with $r$, thereby admitting non-uniform lower bounds. This generalization yields a broader and more flexible notion of convexity that extends beyond classical log-concavity, making Definition 10 substantially weaker than log-concavity (Conforti, 2023; 2024; Conforti et al., 2023b;a; Silveri & Ocello, 2025).*

We modify Conforti et al. (2023b, Lemma 5.9) to our setting, namely when $\beta > 0$[7] to have an explicit expression of the weak convexity constant at each $t \in (0, T)$.

**Lemma 12.** *(Conforti et al., 2023b, Modification of Lemma 5.9) Assume that $U$ is weakly convex as in Definition 10 and fix $t \in (0, T)$. Then, the function $x \mapsto -\log p_t(x)$ is weakly convex with weak convexity profile $\kappa_{-\log p_t(x)}$ satisfying*

$$\kappa_{-\log p_t}(r) \geq \frac{\beta}{\beta + (1-\beta)e^{-2t}} - \frac{e^{-t}}{\beta + (1-\beta)e^{-2t}} \frac{1}{r} f_L \left( \frac{e^{-t}}{\beta + (1-\beta)e^{-2t}} r \right).$$

*In particular, the score function satisfies*

$$\langle \nabla \log p_t(x) - \nabla \log p_t(\bar{x}), x - \bar{x} \rangle \leq -\widehat{C}_t |x - \bar{x}|^2, \quad \text{for } x, \bar{x} \in \mathbb{R}^d, \tag{14}$$

*with*

$$\widehat{C}_t = \frac{\beta}{\beta + (1-\beta)e^{-2t}} - \frac{e^{-2t}}{(\beta + (1-\beta)e^{-2t})^2} L. \tag{15}$$

---

[6]Recently, this notion has been used in the context of score-based generative models in Silveri & Ocello (2025, Definition 3.1).

[7]See Silveri & Ocello (2025, Lemma B.4) for a similar statement.

We show that Assumption 2-(ii) and Assumption 2-(iii) are related to the notion of weak convexity (Definition 10) in the sense made precise in Proposition 13 below. An overview of the proof of Proposition 13 below can be found in Appendix B.

**Proposition 13.** *Let the data distribution $\pi_D$ be in Assumption 2, and let $f_L$ and $L > 0$ be as in Definition 10. Then the potential $U$ is weakly convex as in Definition 10 with*

$$\kappa_U(r) \geq \mu - r^{-1} f_L(r), \quad \text{for all } r > 0, \tag{16}$$

*where $\mu > 0$ in 16 is the strong convexity at infinity constant from Assumption 2-(iii). Conversely, if $U$ is weakly convex as in Definition 10 with lower bound 16 for some known constants $\mu$ and $L > 0$, then*

1. *The potential $U$ is $\widetilde{\mu}$-strongly convex at infinity with $\widetilde{\mu} := \mu - R^{-1} f_L(R) > 0$[8], such that for all $x, \bar{x} \in \mathbb{R}^d$, we have*

$$\langle h(x) - h(\bar{x}), x - \bar{x} \rangle \geq \widetilde{\mu} |x - \bar{x}|^2, \qquad \text{when} \quad |x - \bar{x}| \geq R, \tag{17}$$

   *which holds for all $R > 0$ when $\mu > L$ and for $R \geq R_0 = \frac{2z_0}{L^{1/2}}$ with $z_0$ being the solution of 45 when $\mu \leq L$.*

2. *The potential $U$ is $K$-semiconvex, such that there exists $K \geq 0$ for all $x, \bar{x} \in \mathbb{R}^d$,*

$$\langle h(x) - h(\bar{x}), x - \bar{x} \rangle \geq -K |x - \bar{x}|^2, \qquad \text{when} \quad |x - \bar{x}| < R, \tag{18}$$

   *where $R$ is the same as in point (2).*

As a consequence of Proposition 13 and Lemma 12, one obtains the explicit form of $\widehat{C}_t$ in 14 in our setting, which is given in the following corollary.

**Corollary 14.** *Let $U$ be $K$-semiconvex as in Assumption 2-(ii) and be $\mu$-strongly convex at infinity as in Assumption 2-(iii) and fix $t \in (0, T]$. Then*

$$\langle \nabla \log p_t(x) - \nabla \log p_t(\bar{x}), x - \bar{x} \rangle \leq -\beta_t^{OS} |x - \bar{x}|^2, \qquad \text{for} \quad x, \bar{x} \in \mathbb{R}^d, \tag{19}$$

*where*

$$\beta_t^{OS} = \frac{\mu}{\mu + (1-\mu)e^{-2t}} - \frac{e^{-2t}}{(\mu + (1-\mu)e^{-2t})^2} L, \tag{20}$$

*for some $L > 0$ satisfying 16.*

**Remark 15.** *By Corollary 14 and the proof of Proposition 13, we have*

$$\lim_{t \to 0} \beta_t^{OS} = \mu - L < -K, \tag{21}$$

*which shows that $-\beta_t^{OS}$ is not the lowest bound for the left-hand side of 19. We emphasize that the gap between the limit on the left-hand side of 21 and the semiconvexity constant $K$ is due to the particular choice of $f_L$ in 13 in Proposition 13. This gap may vanish if we replace $f_L$ with an appropriate function $f \in \widetilde{\mathcal{F}}$ (Conforti et al., 2023a, Section 2.1.2), (Conforti et al., 2023b, Section 5.3.1), where*

$$\widetilde{\mathcal{F}} := \left\{ f \in C^2((0, \infty), \mathbb{R}_+) : \ r \mapsto r^{1/2} f(r^{1/2}), \ \text{non-decreasing, concave, bounded such that} \right.$$

$$\left. \lim_{r \downarrow 0} r f(r) = 0, \ f' \geq 0, \quad 2f'' + f f' \leq 0 \right\}.$$

*Note that $\widetilde{\mathcal{F}}$ is non-empty and contains $r \mapsto 2 \tanh(r/2)$. For this reason, we use the constant $K + \mu$ as a proxy of the constant $L$ and replace 20 with the following monotonicity bound*

$$\beta_t^{OS,K,\mu} = \frac{\mu}{\mu + (1-\mu)e^{-2t}} - \frac{e^{-2t}}{(\mu + (1-\mu)e^{-2t})^2} (K + \mu). \tag{22}$$

*Moreover, it holds that*

$$\lim_{t \to 0} \beta_t^{OS,K,\mu} = -K[9],$$

---

[8]We refer to the proof of Proposition 13 in Appendix B below for the derivation of this constant.

[9]Indeed, this shows that $-\beta_0^{OS,K,\mu} < -\beta_0^{OS}$ for the right-hand side of 19 in Corollary 14.

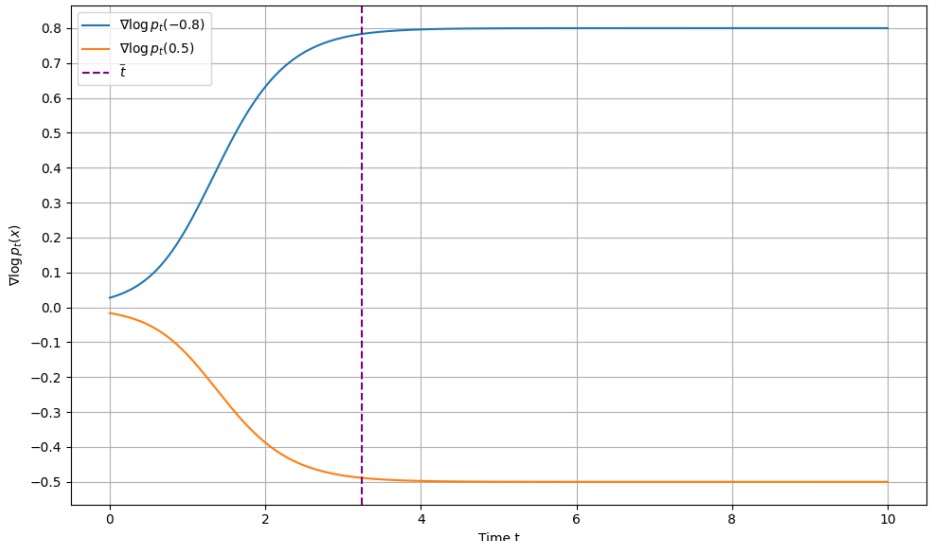

Figure 1: Score function 25 for fixed $x$ and time $\bar{t}$.

*and*

$$\lim_{t\to\infty} \beta_t^{OS} = \lim_{t\to\infty} \beta_t^{OS,K,\mu} = 1,$$

*which is consistent with $\pi_\infty \sim \mathcal{N}(0, I_d)$, the invariant distribution of the OU process.*

Using the explicit expression of 22, we are able to find a time for which the integral of the monotonicity bound[10] $\beta_t^{\text{OS},K,\mu}$ is positive. The proof of the following result is postponed to Appendix B.

**Proposition 16.** *Let $\mu > 0$ and $K \geq 0$. The time integral of $\beta_t^{OS,K,\mu}$ from Remark 15 is*

$$
\begin{aligned}
B(t, 0, \mu, K) &= \int_0^t \left( \frac{\mu}{\mu + (1-\mu)e^{-2s}} - \frac{e^{-2s}}{(\mu + (1-\mu)e^{-2s})^2}(K+\mu) \right) \, \mathrm{d}s \\
&= \frac{1}{2} \left[ \log\left(\mu(e^{2t}-1)+1\right) + \left(\frac{K}{\mu}+1\right)\left(\frac{1}{\mu(e^{2t}-1)+1}-1\right) \right] > 0,
\end{aligned}
\tag{23}
$$

*when $t > t^\star > \bar{t} := \ln\left(\sqrt{1 + \frac{K}{\mu^2}}\right)$ with $t^\star := \inf\{t > 0 : B(t, 0, \mu, K) > 0\}$.*

**Remark 17.** *If we consider the case when $K = 0$ in Assumption 2-(ii), then 23 is satisfied for all $t > 0$.*

**Remark 18.** *We provide a numerical illustration of the critical time of Proposition 23 when a non-log-concave distribution becomes log-concave in the case when $\pi_D$ is a one-dimensional Gaussian mixture with two equi-weighted modes $\eta = 2$, each mode having same variance $s^2 = 9$, namely*

$$\pi_D(\mathrm{d}x) = (2(18\pi)^{1/2})^{-1}\left(\exp\left(-\frac{|x-2|^2}{18}\right) + \exp\left(-\frac{|x+2|^2}{18}\right)\right)\mathrm{d}x. \tag{24}$$

*For this choice of $\pi_D$, the semiconvexity constant $K = \frac{2\eta^2}{(s^2)^2} = \frac{8}{81}$, and the strong convexity at infinity constant $\mu = \frac{s^2 - 2\eta^2}{(s^2)^2} = \frac{1}{81}$. Here, $R \geq 0$. The score function of 24 is given by*

$$\nabla \log p_t(x) = -\frac{x}{9m_t^2 + \sigma_t^2} + \frac{2m_t}{9m_t^2 + \sigma_t^2}\frac{\exp\left\{\frac{-(x-2m_t)^2}{2(9m_t^2+\sigma_t^2)}\right\} - \exp\left\{\frac{-(x+2m_t)^2}{2(9m_t^2+\sigma_t^2)}\right\}}{\exp\left\{\frac{-(x-2m_t)^2}{2(9m_t^2+\sigma_t^2)}\right\} + \exp\left\{\frac{-(x+2m_t)^2}{2(9m_t^2+\sigma_t^2)}\right\}}, \quad t \in [0,T], \; x \in \mathbb{R}, \tag{25}$$

---

[10]Note that $\beta_t^{\text{OS},K,\mu}$ is a function of time.

where $m_t = e^{-t}$ and $\sigma_t^2 = 1 - e^{-2t}$ comes from the representation of the OU process in 52. Figure 1 displays the time behaviour of score function 25 over the time interval $[0, 10]$ for fixed values $x = -0.8$ and $x = 0.5$. We also indicate the time $\bar{t} = \ln\left(\sqrt{1 + \frac{K}{\mu^2}}\right) \approx 3.2377 < t^\star$ from Proposition 25. The score function in Figure 1 converges to $\nabla \log(\pi_\infty(x)) = -x$, where $\pi_\infty = \mathcal{N}(0, 1)$.

### 3.3 Main Results - Optimal Data Dimensional Dependence and Rate of Convergence

The main results are stated as follows. An overview of their proofs can be found in Appendix C.

**Theorem 19.** *Let Assumptions 1, 2, 3.a and 4 hold. Then, there exist constants $C_1$, $C_2$, $C_3$ and $C_4 > 0$ such that for any $T > 0$ and $\gamma, \epsilon \in (0, 1)$,*

$$W_2(\mathcal{L}(Y_J^{EM}), \pi_{\mathsf{D}}) \leq C_1\sqrt{\epsilon} + C_2 e^{-2\int_\epsilon^T \beta_t^{OS,K,\mu} \, \mathrm{d}t - \epsilon} + C_3(T, \epsilon)\sqrt{\varepsilon_{SN}} + C_4(T, \epsilon)\gamma^{1/2}, \tag{26}$$

*where $C_1$, $C_2$, $C_3$ and $C_4$ are given explicitly in Table 3 (Appendix E), $\beta_t^{OS,K,\mu}$ is defined in 22, and its integral is computed in Proposition 16. In addition, the result in 26 implies that for any $\delta > 0$, if we choose $0 < \epsilon < \epsilon_\delta$, $T > T_\delta$, $0 < \varepsilon_{SN} < \varepsilon_{SN,\delta}$ and $0 < \gamma < \gamma_\delta$ with $\epsilon_\delta$, $T_\delta$, $\varepsilon_{SN,\delta}$, and $\gamma_\delta$ given in Table 3, then*

$$W_2(\mathcal{L}(Y_J^{EM}), \pi_{\mathsf{D}}) < \delta.$$

**Remark 20.** *The constant $C_4(T, \epsilon)$ in the error bound in 26 contains the optimal dependence of the data dimension, i.e. $O(\sqrt{d})$, which has been found under the more strict assumption of strong-log concavity of $\pi_{\mathsf{D}}$ in Bruno et al. (2025, Theorem 1 and Remark 12). However, the optimal dependence of the dimension is achieved at the expenses of a worst rate of covergence of order $1/2$.*

The optimal rate of convergence of order $\alpha \in [\frac{1}{2}, 1]$ for the Euler or Milstein scheme of SDEs with constant diffusion coefficients can be attained in Theorem 19 provided that $\mathbb{E}[|\hat{\theta}|^4] < \infty$ and that Assumption 3.a is replaced by Assumption 3.b, as stated in Theorem 21 below.

**Theorem 21.** *Let Assumptions 1, 2, 3.b and 4 hold, and assume that $\mathbb{E}[|\hat{\theta}|^4] < \infty$ Then, there exist constants $C_1$, $C_2$, $C_3$ and $\widetilde{C}_4 > 0$ such that for any $T > 0$ and $\gamma, \epsilon \in (0, 1)$,*

$$W_2(\mathcal{L}(Y_J^{EM}), \pi_{\mathsf{D}}) \leq C_1\sqrt{\epsilon} + C_2 e^{-2\int_\epsilon^T \beta_t^{OS,K,\mu} \, \mathrm{d}t - \epsilon} + C_3(T, \epsilon)\sqrt{\varepsilon_{SN}} + \widetilde{C}_4(T, \epsilon)\gamma^\alpha, \tag{27}$$

*where $C_1$, $C_2$, $C_3$ and $\widetilde{C}_4$ are given explicitly in Table 3 (Appendix E), $\beta_t^{OS,K,\mu}$ is defined in 22, and its integral is computed in Proposition 16. In addition, the result in 27 implies that for any $\delta > 0$, if we choose $0 < \epsilon < \epsilon_\delta$, $T > T_\delta$, $0 < \varepsilon_{SN} < \varepsilon_{SN,\delta}$ and $0 < \gamma < \widetilde{\gamma}_\delta$ with $\epsilon_\delta$, $T_\delta$, $\varepsilon_{SN,\delta}$, and $\widetilde{\gamma}_\delta$ given in Table 3, then*

$$W_2(\mathcal{L}(Y_J^{EM}), \pi_{\mathsf{D}}) < \delta. \tag{28}$$

**Remark 22.** *The constant $\widetilde{C}_4$, explicitly given in Table 3 (Appendix E) exhibits a linear dependence on the data dimension, i.e., $O(d)$. This scaling arises from the explicit Milstein scheme developed in Kumar & Sabanis (2019), which relies on Assumption 3.b and is leveraged in the proof of Theorem 21 to achieve the optimal convergence rate of order $\alpha \in [\frac{1}{2}, 1]$. This explicit Milstein scheme requires control on the fourth moment of the one-step discretization, see for instance, Lemma 25 (Appendix C) which enables a convergence rate in $W_2$ consistent with the known optimal rate of convergence for the Euler or Milstein scheme of SDEs with constant diffusion coefficients. However, this comes at the cost of a worse dependence on the data dimension.*

### 3.4 Examples of potentials satisfying by Assumption 2

We present several examples to demonstrate the wide applicability of our Assumption 2 to a broad class of data distributions, some of which are not covered by previous results in Wasserstein distance of order two (Silveri & Ocello, 2025; Strasman et al., 2025; Gao et al., 2025; Bruno et al., 2025; Tang & Zhao, 2024; Yu & Yu, 2025).

### 3.4.1 Symmetric modified half-normal distribution

We consider the case of a one-dimensional symmetric modified half-normal distribution

$$\pi_{\mathsf{D}}(\mathrm{d}x) = \frac{\sqrt{\xi}\exp\left(-\xi x^2 - |x|\right)}{\Psi\left(\frac{1}{2}, \frac{-1}{\sqrt{\xi}}\right)}\,\mathrm{d}x, \quad x \in \mathbb{R}, \tag{29}$$

for some unknown $\xi > 0$ and normalizing constant

$$\Psi\left(\frac{1}{2}, \frac{-1}{\sqrt{\xi}}\right) := \sum_{n=0}^{\infty} \frac{\Gamma\left(\frac{1}{2} + \frac{n}{2}\right)}{\Gamma(n)} \frac{(-1)^n \xi^{-n/2}}{n!},$$

where $\Gamma(n)$ is the Gamma function. We refer the reader to Appendix D for additional details about the derivation of 29. As highlighted in Sun et al. (2023, Section 2), the modified half-normal distribution appears in several Bayesian statistical methods as a posterior distribution to sample from in Bayesian Binary regression, analysis of directional data, and Bayesian graphical models.

Assumption 2-(i) is satisfied for $U(x) = \xi x^2 + |x|$. In addition, we have, for all $x, \bar{x} \in \mathbb{R}$

$$\begin{aligned}\langle h(x) - h(\bar{x}), x - \bar{x}\rangle &= 2\left(\xi|x - \bar{x}|^2 + (x - \bar{x})\mathbb{1}_{x>0,\ \bar{x}<0} - (x - \bar{x})\mathbb{1}_{x<0,\ \bar{x}>0}\right) \\ &\geq 2\xi|x - \bar{x}|^2,\end{aligned} \tag{30}$$

which shows that Assumption 2-(ii) is verified for any $K \geq 0$, and Assumption 2-(iii) is verified for $\mu = 2\xi$. Therefore, we can conclude that 29 satisfies Assumption 2.

### 3.4.2 Multidimensional Gaussian mixture distribution

We consider a multidimensional Gaussian mixture data distribution with unknown mean and variance, i.e.,

$$\pi_{\mathsf{D}}(\mathrm{d}x) = \sum_{i=1}^{I} \widetilde{\xi}_i \frac{1}{(2\pi s_i^2)^{d/2}} \exp\left(-\frac{|x - \eta_i|^2}{2s_i^2}\right)\,\mathrm{d}x, \quad x \in \mathbb{R}^d, \tag{31}$$

with $s_i > 0$, $\eta_i \in \mathbb{R}^d$, and $\widetilde{\xi}_i \in [0, 1]$ for $i \in \{1, \dots, I\}$ such that $\sum_{i=1}^{J} \widetilde{\xi}_i = 1$. The authors in Silveri & Ocello (2025, Appendix A) show that the score function of 31 is Lipschitz continuous and $-\log \pi_{\mathsf{D}}$ is weakly convex. Therefore, Assumption 2 is satisfied. In addition, the distribution 31 covers also case of the double-well potential:

$$U(x) = |x|^4 - |x|^2, \quad x \in \mathbb{R}^d, \tag{32}$$

which is 2-semiconvex and strongly convex at infinity.

### 3.4.3 Multi-dimensional Potentials

Similarly as in Section 3.4.1, one can proves that the elastic net potential:

$$U(x) = |x|^2 + \sum_{i=1}^{d} |x_i|, \quad x \in \mathbb{R}^d, \tag{33}$$

satisfies Assumption 2. Moreover, the following potential

$$U(x) = \max\left\{|x|, |x|^2\right\}, \qquad x \in \mathbb{R}^d, \tag{34}$$

verifies Assumption 2 with $K = 0$, $R = 1$, and $\mu = 2$ as well as the following non-convex potential presented in Johnston et al. (2025, Example 4.2):

$$U(x) = \max\left\{|x|, |x|^2\right\} - \frac{1}{2}|x|^2, \qquad x \in \mathbb{R}^d. \tag{35}$$

# 4   Related Work and Comparison

In recent years, there has been a rapidly expanding body of research on the convergence theory of Score-based Generative Models. Existing works for convergence bounds can be divided into two main approaches, depending on the divergence or distance used.

The first approach focuses on $\alpha$-divergences, particularly the Kullback–Leibler (KL) divergence and Total Variation (TV) distance (e.g., Benton et al. (2024); Conforti et al. (2025); Yang & Wibisono (2023); Li & Cai (2024); Block et al. (2020); De Bortoli et al. (2021); Lee et al. (2022); Li et al. (2024); Lee et al. (2023); Chen et al. (2023a;b); Oko et al. (2023); Liang et al. (2025); Yang & Wibisono (2022)), which are the vast majority of the results available in the literature. Crucially, bounds on KL divergence imply bounds on TV distance via Pinsker's inequality, strengthening their wide applicability. We provide a brief and selective overview of some of the findings following this first approach. The results in TV distance in Lee et al. (2022) and in KL divergence Yang & Wibisono (2023) established convergence bounds characterized by polynomial complexity under the assumption that the data distribution satisfies a logarithmic Sobolev inequality and that the score function is Lipschitz continuous. By replacing the requirement that the data distribution satisfies a functional inequality with the assumption that $\pi_\mathrm{D}$ has finite KL divergence with respect to the standard Gaussian and by assuming that the score function for the forward process is Lipschitz, the authors in Chen et al. (2023b) managed to derive bounds in TV distance which scale polynomially in all the problem parameters. By requiring only the Lipschitzness of the score at the initial time rather than along the full trajectory, the authors in Chen et al. (2023a, Theorem 2.5) managed to establish, using an exponentially decreasing then linear step size, convergence bounds in KL divergence with quadratic dimensional dependence and logarithmic complexity in the Lipschitz constant. Later, Benton et al. (2024) provided KL convergence bounds that are linear in the data dimension, up to logarithmic factors, by assuming finite second moments of the data distribution and employing early stopping. However, both the results of Chen et al. (2023a, Theorem 2.5) and Benton et al. (2024, Theorem 1 and Corollary 1) still require the uniqueness of solutions for the backward SDE 2, and therefore additional assumptions on the score function are needed. For further discussion on this point, we refer the reader to Bruno et al. (2025, Section 4.2). Assuming finite second moments and using an exponential integrator (EI) scheme with both constant and exponentially decaying step sizes, the authors in Conforti et al. (2025, Corollary 2.4) derive a KL divergence bound with early stopping, which scales linearly in the data dimension up to logarithmic factors. Bounds in KL without early stopping have been derived in Conforti et al. (2025) for data distributions with finite Fisher information with respect to the standard Gaussian distribution. We note that this condition on $\pi_\mathrm{D}$ stated in Conforti et al. (2025, Assumption H2) still requires that the potential $U \in C^1(\mathbb{R}^d)$. The KL bounds provided in Conforti et al. (2025, Theorem 2.1 and 2.2) scale linearly in the Fisher information when an EI discretization scheme with constant step size is used, and logarithmically in the Fisher information when an exponential-then-constant step size Conforti et al. (2025, Theorem 2.3) is employed.

The second approach focuses on convergence bounds in Wasserstein distance, a metric which is often considered more practical and informative for estimation tasks. We can relate results following this approach with the results of the first approach only when $\pi_\mathrm{D}$ is a strongly log-concave distribution. In this case, $W_2$-bounds in terms of KL divergence follow from an extension of Talagrand's inequality (Gozlan & Léonard, 2010, Corollary 7.2). However, for two general data distributions, there is no known relationship between their KL divergence and their $W_2$. Therefore, we cannot compare our findings in Theorem 19 and Theorem 21 with the results derived following the first approach. One line of work within the second approach assumes (at least) strong log-concavity of the data distribution (Strasman et al., 2025; Gao et al., 2025; Bruno et al., 2025; Tang & Zhao, 2024; Yu & Yu, 2025). Under this (strict) assumption, Bruno et al. (2025, Remark 12) achieved optimal data dimensional dependence, i.e., reaching $O(\sqrt{d})$. The recent bound in Silveri & Ocello (2025, Theorem D.1) exhibits similar scaling in $d$ while relaxing the strong log-concavity assumption on $\pi_\mathrm{D}$ to weakly log-concavity, but still requiring that the potential $\nabla^2 U$ exists (see, e.g., Silveri & Ocello (2025, Proof of Proposition B.1 and B.2)). Our Assumption 2 is much weaker than this requirement and it allows to consider the case of potentials with discontinuous gradients covering a wider range of distributions as outlined in Section 3.4. Another line of work following this approach focuses on specific structural assumptions of the data distribution. For instance, convergence bounds in Wasserstein distance of order one with exponential dependence on the problem parameters have been obtained in Bortoli (2022) under the so-called manifold

hypothesis, namely assuming that the target distribution is supported on a lower-dimensional manifold or is given by some empirical distribution. Under the same metric, the authors in Mimikos-Stamatopoulos et al. (2024) provide a convergence analysis when the data distribution is defined on a torus. Under the $W_2$ metric, Wang & Wang (2024) derive convergence bounds assuming that the tail of $\pi_D$ is Gaussian and that $U \in C^2$, which is a stronger condition than merely requiring $\nabla U$ to be Lipschitz. We summarize in Table 1 and the best results obtained in $W_2$, i.e., Bruno et al. (2025); Silveri & Ocello (2025) and compare with our best result, which scale polynomially in the data dimension, i.e. $O(\sqrt{d})$ in Theorem 19. As mentioned in the Introduction, previous $W_2$ bounds (Silveri & Ocello, 2025; Gao et al., 2025; Strasman et al., 2025; Tang & Zhao, 2024) require the stepsize $\gamma$ of the generative algorithm to be tuned based on quantities that are often difficult to compute in practice, such as the Lipschitz or strong convexity constant of the data distribution, which can lead to very small stepsizes. In contrast, Theorem 19 and Theorem 21 impose no such restrictions, making them more suitable for practical use. Table 2 summarizes the assumptions on $\gamma$ in prior works and compares them with ours.

We close this section by briefly commenting on the choice of deriving our results in Wasserstein distance of order two. Beyond its theoretical relevance, this choice is motivated by practical considerations in generative modeling. First, the Wasserstein distance is often regarded as a more informative and robust metric for estimation tasks. Second, a widely used performance metric for evaluating the quality of images produced by generative models is the Fréchet Inception Distance (FID) Heusel et al. (2017), which measures the Fréchet distance between the distributions of generated and real samples, assuming Gaussian distributions. In particular, this Fréchet distance is equivalent to the Wasserstein-2 distance. Thus, providing convergence results under the Wasserstein-2 metric enhances the practical relevance of our theoretical findings.

Table 1: Summary of previous bounds for $W_2(\mathcal{L}(\widehat{Y}_J^{\mathrm{EM}}), \pi_D)$ and our result in Theorem 19. All the bounds assume that $\pi_D(\mathrm{d}x) \propto e^{-U(x)}\mathrm{d}x$ has finite second moments.

| Assumption on $\pi_D$ | Error bound | Reference |
|---|---|---|
| $U$ strongly convex, $\nabla \log p_t(0) \in L^2([\epsilon,T])$, and Assumption 4 | $O(\sqrt{d})\sqrt{\epsilon} + O(\sqrt{d})e^{-2\widehat{L}_{\mathrm{MO}}(T-\epsilon)-\epsilon} + O(e^{(1+\zeta-2\widehat{L}_{\mathrm{MO}})(T-\epsilon)})\sqrt{\varepsilon_{\mathrm{SN}}} + O(\sqrt{d}e^{T^{2\alpha+1}}T^{2\alpha+1}\widetilde{\varepsilon}_{\mathrm{AL}}^{1/2})\gamma^{1/2}$, 

 with $\widehat{L}_{\mathrm{MO}} > 0$ lower bound of the strongly convex constant of $U$, see e.g., Bruno et al. (2025, Remark 4). | Bruno et al. (2025, Remark 12) |
| $U \in C^2(\mathbb{R}^d)$, weakly convex, and Assumption 4 | $e^{(2L_U+5)\eta(\beta,L,(2L_U+5)^2\gamma/2)}[e^{-T}W_2(\pi_D,\pi_\infty) + 4\varepsilon_{\mathrm{SN}}(T - \eta(\beta,L,0)) + \sqrt{2\gamma}(4L_U\sqrt{d} + 6\sqrt{d} + \sqrt{d+\mathbb{E}[|X_0|^2]})(T-\eta(\beta,L,0))]$, 

 with $L_U \geq 0$ one-sided Lipschitz constant for $\nabla U$, see e.g., Silveri & Ocello (2025, Assumption H1), $\eta(\beta,L,\gamma)$ defined in (Silveri & Ocello, 2025, equation (29)), and $\gamma < 2/(2L_U + 5)^2$ . | Silveri & Ocello (2025, Theorem D.1) |
| Assumption 2 and Assumption 4 | $$O(\sqrt{d})\sqrt{\epsilon} + O(\sqrt{d})e^{-2\int_\epsilon^T \beta_t^{\mathrm{OS},K,\mu}\,\mathrm{d}t - \epsilon}$$ $$+ O(e^{(1+\zeta)(T-\epsilon)-2\int_\epsilon^T \beta_t^{\mathrm{OS},K,\mu}\,\mathrm{d}t})\sqrt{\varepsilon_{\mathrm{SN}}} + O(\sqrt{d}e^{T^{2\alpha+1}}T^{3\alpha+1}\widetilde{\varepsilon}_{\mathrm{AL}}^{1/2})\gamma^{1/2}.$$ | Theorem 19 |

Table 2: Summary of restrictions on the stepsizes of the generative algorithm $\widehat{Y}_J^{\mathrm{EM}}$ used in the previous bounds for $W_2(\mathcal{L}(\widehat{Y}_J^{\mathrm{EM}}), \pi_{\mathsf{D}})$ and our results in Theorem 19 and Theorem 21. All the bounds assume that $\pi_{\mathsf{D}}(\mathrm{d}x) \propto e^{-U(x)}\mathrm{d}x$ has finite second moments.

| Assumption on $\pi_{\mathsf{D}}$ | Restriction on the stepsize | Reference |
|---|---|---|
| $U$ strongly convex, $\nabla U$ Lipschitz continuous, and Assumption 4 | $0 < \gamma \le \min_{0 \le t \le T} \left( \dfrac{1 - e^{-2t}(\frac{1}{m_0} - 1)}{(1 + e^{-2t}(\frac{1}{m_0} - 1))(1 + 4(\widetilde{L}(t))^2 + 2M_1)} \right)$ and $0 < \gamma \le \min_{0 \le t \le T} \left( \dfrac{1 + (\frac{1}{m_0} - 1)e^{-2t}}{1 - (\frac{1}{m_0} - 1)e^{-2t}} \right),$ with $m_0 > 0$ strong convexity constant for $U$, see e.g., Gao et al. (2025, Assumption 1), $M_1 > 0$ defined in Gao et al. (2025, Assumption 2), $\widetilde{L}(t) = \min_{0 \le t \le T}((1 - e^{-2t})^{-1}, e^{2t}L_0)$ Lipschitz constant of $\nabla \log p_t(x)$, and $L_0 > 0$ Lipschitz constant for $\nabla U$, see e.g., Gao et al. (2025, Assumption 1). | Gao et al. (2025, Assumption 4) used in Gao et al. (2025, Theorem 2) |
| $U$ strongly convex, $\nabla \log p_t(0) \in L^2([\epsilon, T])$, and Assumption 4 | $\gamma \in (0, 1).$ | Bruno et al. (2025, Remark 12) |
| $U$ strongly convex, and $\nabla U$ Lipschitz continuous | $0 < \gamma < \dfrac{C(T - t)}{\left( \max_{t_j \le s \le t_{j+1}} L(T - s) \right) L(T - t)} e^{-(t_{j+1} - t_j)},$ with $\{t_j,\ 0 \le j \le J\}$ regular discretization of $[0, T]$, $C_{T-t}$ and $L_{T-t}$ are strong log-concavity and Lipschitz constant, respectively for $\nabla \log \frac{p_t}{\varphi_{\sigma^2}}$, where $\varphi_{\sigma^2}$ is the density function of a mean zero Gaussian distribution with variance $\sigma^2 I_d$. | Strasman et al. (2025, Proposition C.3) used in Strasman et al. (2025, Theorem 4.2) |
| $U$ strongly convex, $\nabla U$ Lipschitz continuous, and Assumption 4 | $0 < \gamma < \min \left( \dfrac{1}{2}, \dfrac{\kappa}{2T(1 + \kappa)} \right),$ with $\kappa > 0$ strong convexity constant. | Tang & Zhao (2024, Theorem 3) |
| $U \in C^2(\mathbb{R}^d)$, weakly convex, and Assumption 4 | $0 < \gamma < \dfrac{2}{(2L_U + 5)^2},$ with $L_U \ge 0$ one-sided Lipschitz constant for $\nabla U$, see e.g., Silveri & Ocello (2025, Assumption H1). | Silveri & Ocello (2025, Theorem D.1) |
| Assumption 2, and Assumption 4 | $\gamma \in (0, 1).$ | Theorem 19, and Theorem 21. |

**Acknowledgments.**

This work was supported by Innovate UK [grant number 10081810]. This work has received funding from the Ministry of Trade, Industry and Energy (MOTIE) and Korea Institute for Advancement of Technology (KIAT) through the International Cooperative R&D program (No.P0025828). This work has been partially

supported by project MIS 5154714 of the National Recovery and Resilience Plan Greece 2.0 funded by the European Union under the NextGenerationEU Program.

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

## Appendix

## A Regularity of the Score Function

We recall the following result to justify the smoothness of the map

$$(0, T] \times \mathbb{R}^d \ni (t, x) \mapsto p_t(x) \in \mathbb{R}_+, \tag{36}$$

where $p_t$ density of the forward process defined in Section 2.

**Proposition 23.** *(Conforti et al., 2025, Proposition 3.1) Let $\pi_D$ be absolutely continuous with respect to the Lebesgue measure, and denote its density by $p_0$. The map defined in 36 is positive and solution of the following Fokker–Planck equation on $(0, T] \times \mathbb{R}^d$:*

$$\partial_t p_t(x) - div(x \; p_t) - \Delta p_t(x) = 0, \quad for \; (t, x) \in (0, T] \times \mathbb{R}^d.$$

*Moreover, it belongs to $C^{1,2}((0, T] \times \mathbb{R}^d)$; i.e. for any $t \in (0, T)$, $x \mapsto p_t(x)$ is twice continuously differentiable, and for any $x \in \mathbb{R}^d$, $t \mapsto p_t(x)$ is continuously differentiable on $(0, T]$.*

## B Further Details on Assumption 2 and Weak Convexity of the Data Distribution

We provide the proofs of Section 3.2.

*Proof of Proposition 13.* We begin by considering that $\pi_D$ satisfies Assumption 2. Recall that $f_L$ is defined as in 13. Note that $r \mapsto r^{-1} f_L(r)$ is non-increasing on $(0, \infty)$ and $f'_L(0) = L > r^{-1} f_L(r)$ for $r \in (0, R]$. We look for $L > 0$ satisfying

$$\inf_{r \in (0, R]} r^{-1} f_L(r) = R^{-1} f_L(R) = 2R^{-1} L^{1/2} \tanh((RL^{1/2})/2) = K + \mu. \tag{37}$$

Equivalently, we look for $x = L^{1/2} R/2 > 0$ such that

$$x \tanh(x) = \frac{K + \mu}{4} R^2, \quad \text{subject to} \quad x > \frac{\sqrt{K + \mu}}{2} R, \tag{38}$$

so as $L > K + \mu$. Note that $\tanh(x) \leq x$ for all $x \geq 0$. Therefore, if we choose $x = \frac{\sqrt{K+\mu}}{2}R$, then

$$\frac{\sqrt{K+\mu}}{2}R \tanh\left(\frac{\sqrt{K+\mu}}{2}R\right) \leq \frac{K+\mu}{4}R^2. \tag{39}$$

Using 39 and $\lim_{x\uparrow\infty} x\tanh(x) = \infty$, we deduce that there exists $x^\star > 0$ such that

$$x^\star \tanh(x^\star) = \frac{K+\mu}{4}R^2, \tag{40}$$

with $x^\star > \frac{\sqrt{K+\mu}}{2}R$, since $x \mapsto x\tanh(x)$ is non-decreasing on $(0,\infty)$. By Assumption 2 and 37, we have

$$\begin{aligned} k_U(r) &\geq \mu - (K+\mu) \\ &\geq \mu - r^{-1}f_L(r), \qquad \text{for } r \leq R. \end{aligned} \tag{41}$$

Moreover,

$$\begin{aligned} k_U(r) &\geq \mu \\ &\geq \mu - r^{-1}f_L(r), \qquad \text{for } r > R, \end{aligned}$$

where it is used that $r^{-1}f_L(r) > 0$ for all $r > 0$. This proves the first part of the statement in Proposition 13, i.e. the lower bound 16.

Conversely, assume that $U$ is weakly convex as in Definition 10 with lower bound 16 for some known constants $\mu$ and $L > 0$. We look for $R$ such that

$$\begin{aligned} \kappa_U(r) &\geq \mu - r^{-1}f_L(r) \\ &\geq \mu - R^{-1}f_L(R) \\ &> 0, \qquad\qquad \forall\, r > R, \end{aligned} \tag{42}$$

where it is used that $r^{-1}f_L(r)$ is decreasing on $(0,\infty)$. Let $\widetilde{\mu} := \mu - R^{-1}f_L(R)$, so 42 becomes $\kappa_U(r) \geq \widetilde{\mu} > 0$, for all $r > R$. One notes that

$$\widetilde{\mu} = \mu - L\frac{\tanh((RL^{1/2})/2)}{(RL^{1/2})/2} > 0. \tag{43}$$

If $\mu > L$, 43 is satisfied for all $R > 0$. If $\mu \leq L$, 43 holds for $R \geq R_0$, where $R_0$ is the unique solution to

$$\mu = \frac{2L^{1/2}}{R}\tanh\left(\frac{RL^{1/2}}{2}\right). \tag{44}$$

Let $z = \frac{RL^{1/2}}{2}$, then $R_0 = \frac{2z_0}{L^{1/2}}$, where $z_0$ solves

$$\frac{\tanh(z)}{z} = \frac{\mu}{L}. \tag{45}$$

Since $\frac{\tanh(z)}{z}$ monotonically decreases from 1 to 0 as $z$ increases, a unique $z_0 > 0$ solving 45 exists for $\mu < L$. Therefore, 42 is satisfied for $R \geq R_0 = \frac{2z_0}{L^{1/2}}$. This proves that $U$ is $\widetilde{\mu}$-strongly convex at infinity, and therefore 17. Using the assumption that $U$ is weakly convex as in Definition 10, one obtains that

$$\begin{aligned} \kappa_U(r) &\geq \mu - r^{-1}f_L(r) \\ &\geq \mu - L, \qquad \text{for} \quad r \leq R. \end{aligned} \tag{46}$$

We distinguish two cases for the lower bound in 46. If $\mu > L$, then $\kappa_U(r) \geq -K$ for $r \leq R$ for all $R > 0$ and $K \geq 0$. If $\mu \leq L$, then, by setting $K = L - \mu$ in 45, we have $\kappa_U(r) \geq -K$ for $r \leq R$ for all $R > 0$. This proves that $U$ is $K$-semiconvex, and therefore 18. This concludes the proof for the second part of the statement in Proposition 13.

$\square$

*Proof of Proposition 16.* We look for $t^\star$ satisfying

$$B(t^\star, 0, \mu, K) = \frac{1}{2}\left[\log\left(\mu(e^{2t^\star} - 1) + 1\right) + \left(\frac{K}{\mu} + 1\right)\left(\frac{1}{\mu(e^{2t^\star} - 1) + 1} - 1\right)\right] > 0. \tag{47}$$

Equivalently, we look for $x := e^{2t^\star} - 1$ such that

$$g(\mu x + 1) = \log(\mu x + 1) - \left(\frac{K}{\mu} + 1\right)\frac{\mu x}{\mu x + 1} > 0. \tag{48}$$

Note that 48 is satisfied for all $x > 0$ when $K = 0$. In addition, we have

$$\begin{aligned} \lim_{x \to 0+} g(\mu x + 1) = 0, \\ \lim_{x \to +\infty} g(\mu x + 1) = \infty, \end{aligned} \tag{49}$$

and

$$\begin{aligned} \frac{\mathrm{d}}{\mathrm{d}x}g(\mu x + 1) = \frac{\mu}{\mu x + 1} - \frac{K + \mu}{(\mu x + 1)^2} \geq 0 \quad \text{when} \quad x \geq \frac{K}{\mu^2}. \\ \frac{\mathrm{d}^2}{\mathrm{d}x^2}g(\mu x + 1) = -\frac{\mu^2}{(\mu x + 1)^2} + \frac{2(K + \mu)\mu}{(\mu x + 1)^3} \geq 0 \quad \text{when} \quad x \leq \frac{2K}{\mu^2} + \frac{1}{\mu}. \end{aligned} \tag{50}$$

By 50, the function $g$ in 48 has a minimum at $\frac{K}{\mu^2}$ and

$$g\left(\frac{K}{\mu} + 1\right) = \log\left(\frac{K}{\mu} + 1\right) - \frac{K}{\mu} < 0,$$

for all $K, \mu > 0$. By 49 and 50, there exists $x > \frac{K}{\mu^2}$ such that 48 is strictly positive. Therefore, there exists $t^\star > \ln\left(\sqrt{1 + \frac{K}{\mu^2}}\right)$ such that 47 holds. $\qquad\square$

# C   Proof of the Main Results

In this section, we present the proofs of Theorem 19 and Theorem 21. We begin by recalling an upper bound on the moments of the process $(\widehat{Y}_t^{\text{EM}})_{t \in [0, T - \epsilon]}$ defined in 7, along with an estimate for its one-step discretization error. These results will be instrumental in the subsequent proofs.

**Lemma 24.** *(Bruno et al., 2025, Lemma 20) Let Assumptions 1 and 3.a hold, and suppose that $\mathbb{E}[|\hat\theta|^p] < \infty$ for any $p \in [2, 4]$. Then, for any $t \in [0, T - \epsilon]$,*

$$\sup_{0 \leq s \leq t} \mathbb{E}\left[|\widehat{Y}_s^{EM}|^p\right] \leq C_{\text{EM},p}(t),$$

*where*

$$\begin{aligned} C_{\text{EM},p}(t) := {} & e^{t(3p - 1 - \frac{2}{p} + 2^{2p-1}\mathsf{K}_{Total}^p(1 + T^{\alpha p}))} \\ & \times \left(\mathbb{E}\left[|\widehat{Y}_0^{EM}|^p\right] + 2^{3p-2}\mathsf{K}_{Total}^p t(1 + \mathbb{E}[|\hat\theta|^p])(1 + T^{\alpha p}) + \frac{2}{p}(pd + p(p - 2))^{\frac{p}{2}}t\right), \end{aligned}$$

*and $\mathsf{K}_{Total}$ is defined in Remark 8.*

**Lemma 25.** *(Bruno et al., 2025, Lemma 21) Let Assumptions 1 and 3.a hold, and suppose that $\mathbb{E}[|\hat\theta|^p] < \infty$ for any $p \in [2, 4]$. Then, for any $t \in [0, T - \epsilon]$,*

$$\mathbb{E}\left[|\widehat{Y}_t^{EM} - \widehat{Y}_{\lfloor t/\gamma \rfloor \gamma}^{EM}|^p\right] \leq \gamma^{\frac{p}{2}}C_{\text{EMose},p},$$

*where*

$$\begin{aligned} C_{\text{EMose},p} := {} & 2^{p-1}(C_{\text{EM},p}(T) + \mathsf{K}_{Total}^p(1 + T^{\alpha p})(2^{3p-2}C_{\text{EM},p}(T) + 2^{4p-3}(1 + \mathbb{E}[|\hat\theta|^p]))) \\ & + (dp(p - 1))^{\frac{p}{2}}, \end{aligned}$$

*$C_{\text{EM},p}$ and $\mathsf{K}_{Total}$ are defined in Lemma 24 and in Remark 8, respectively.*

*Proof of Theorem 19.* We derive the non-asymptotic estimate for $W_2(\mathcal{L}(Y_J^{\text{EM}}), \pi_{\mathsf{D}})$ using the splitting

$$
\begin{aligned}
W_2(\mathcal{L}(Y_J^{\text{EM}}), \pi_{\mathsf{D}}) \leq{}& W_2(\pi_{\mathsf{D}}, \mathcal{L}(Y_{t_J})) + W_2(\mathcal{L}(Y_{t_J}), \mathcal{L}(\widetilde{Y}_{t_J})) \\
&+ W_2(\mathcal{L}(\widetilde{Y}_{t_J}), \mathcal{L}(Y_{t_J}^{\text{aux}})) + W_2(\mathcal{L}(Y_{t_J}^{\text{aux}}), \mathcal{L}(Y_J^{\text{EM}})).
\end{aligned}
\tag{51}
$$

We provide upper bounds on the error made by the early stopping, i.e. $W_2(\pi_{\mathsf{D}}, \mathcal{L}(Y_{t_J}))$, the error made by approximating the initial condition of the backward process $Y_0 \sim \mathcal{L}(X_T)$ with $\widetilde{Y}_0 \sim \pi_\infty$, i.e. $W_2(\mathcal{L}(Y_{t_J}), \mathcal{L}(\widetilde{Y}_{t_J}))$, the error made by approximating the score function with $s$, i.e. $W_2(\mathcal{L}(\widetilde{Y}_{t_J}), \mathcal{L}(Y_{t_J}^{\text{aux}}))$, and the discretisation error, i.e. $W_2(\mathcal{L}(Y_{t_J}^{\text{aux}}), \mathcal{L}(Y_J^{\text{EM}}))$, separately.

**Upper bound on $W_2(\pi_{\mathsf{D}}, \mathcal{L}(Y_{t_J}))$.** This bound can be established by following the same argument as in (Bruno et al., 2025, Proof of Theorem 10), which relies on the representation of the OU process

$$
X_t \overset{\text{a.s.}}{=} m_t X_0 + \sigma_t Z_t, \quad m_t = e^{-t}, \quad \sigma_t^2 = 1 - e^{-2t}, \quad Z_t \sim \mathcal{N}(0, I_d),
\tag{52}
$$

where $\overset{\text{a.s.}}{=}$ denotes almost sure equality. Therefore, we have

$$
W_2(\pi_{\mathsf{D}}, \mathcal{L}(Y_{t_J})) \leq 2\sqrt{\epsilon}(\sqrt{\mathbb{E}[|X_0|^2]} + \sqrt{d}),
\tag{53}
$$

where $t_J = T - \epsilon$.

**Upper bound on $W_2(\mathcal{L}(Y_{t_J}), \mathcal{L}(\widetilde{Y}_{t_J}))$.** Using Itô's formula, we have, for any $t \in [0, T - \epsilon]$,

$$
\begin{aligned}
\mathrm{d}|Y_t - \widetilde{Y}_t|^2 &= 2\langle Y_t - \widetilde{Y}_t, Y_t + 2\nabla \log p_{T-t}(Y_t) - \widetilde{Y}_t - 2\nabla \log p_{T-t}(\widetilde{Y}_t)\rangle \, \mathrm{d}t \\
&= 2|Y_t - \widetilde{Y}_t|^2 \, \mathrm{d}t + 4\langle Y_t - \widetilde{Y}_t, \nabla \log p_{T-t}(Y_t) - \nabla \log p_{T-t}(\widetilde{Y}_t)\rangle \, \mathrm{d}t.
\end{aligned}
\tag{54}
$$

By integrating and taking on both sides in 54, we have

$$
\begin{aligned}
\mathbb{E}\left[|Y_{t_J} - \widetilde{Y}_{t_J}|^2\right] ={}& \mathbb{E}\left[|Y_0 - \widetilde{Y}_0|^2\right] + \int_0^{t_J} 2\mathbb{E}\left[|Y_t - \widetilde{Y}_t|^2\right] \, \mathrm{d}t \\
&+ \int_0^{t_J} 4\mathbb{E}\left[\langle Y_t - \widetilde{Y}_t, \nabla \log p_{T-t}(Y_t) - \nabla \log p_{T-t}(\widetilde{Y}_t)\rangle\right] \, \mathrm{d}t.
\end{aligned}
\tag{55}
$$

By integrating, taking expectations on both sides in 55, using Corollary 14, the representation 52 with $Z_T \overset{\text{d}}{=} \widetilde{Y}_0$ (where $\overset{\text{d}}{=}$ denotes equality in distribution), the inequality $1 - \sigma_t \leq m_t$ for any $t \in [0, T]$, we have

$$
\begin{aligned}
&\mathbb{E}\left[|Y_{t_J} - \widetilde{Y}_{t_J}|^2\right] \\
&\leq \mathbb{E}\left[|Y_0 - \widetilde{Y}_0|^2\right] + 2\int_0^{t_J} \mathbb{E}\left[|Y_t - \widetilde{Y}_t|^2\right] \, \mathrm{d}t - 4\int_0^{t_J} \beta_{T-t}^{\text{OS}} \mathbb{E}\left[|Y_t - \widetilde{Y}_t|^2\right] \mathrm{d}t \\
&\leq \mathbb{E}[|Y_0 - \widetilde{Y}_0|^2] e^{2[t_J - 2\int_0^{t_J} \beta_{T-t}^{\text{OS}} \, \mathrm{d}t]} \\
&= \mathbb{E}[|m_T X_0 + (\sigma_T - 1)\widetilde{Y}_0|^2] e^{2[t_J - 2\int_0^{t_J} \beta_{T-t}^{\text{OS}} \, \mathrm{d}t]} \\
&\leq 2\left(\mathbb{E}[|X_0|^2] + d\right) e^{2[t_J - 2\int_0^{t_J} \beta_{T-t}^{\text{OS}} \, \mathrm{d}t] - 2T}.
\end{aligned}
\tag{56}
$$

Using 56, Remark 15, and and $t_J = T - \epsilon$, we have

$$
\begin{aligned}
W_2(\mathcal{L}(Y_{t_J}), \mathcal{L}(\widetilde{Y}_{t_J})) &\leq \sqrt{\mathbb{E}[|Y_{t_J} - \widetilde{Y}_{t_J}|^2]} \\
&\leq \sqrt{2}(\sqrt{\mathbb{E}[|X_0|^2]} + \sqrt{d}) e^{-2\int_\epsilon^T \beta_t^{\text{OS}, K, \mu} \, \mathrm{d}t - \epsilon}.
\end{aligned}
\tag{57}
$$

**Upper bound on** $W_2(\mathcal{L}(\widetilde{Y}_{t_J}), \mathcal{L}(Y_{t_J}^{\mathbf{aux}}))$. Using Itô's formula, we have, for $t \in [0, T - \epsilon]$,

$$
\begin{aligned}
\mathrm{d}|\widetilde{Y}_t - Y_t^{\mathrm{aux}}|^2 &= 2\langle \widetilde{Y}_t - Y_t^{\mathrm{aux}}, \widetilde{Y}_t + 2\,\nabla \log p_{T-t}(\widetilde{Y}_t) - Y_t^{\mathrm{aux}} - 2\,s(T-t, \hat{\theta}, Y_t^{\mathrm{aux}})\rangle\,\mathrm{d}t \\
&= 2|\widetilde{Y}_t - Y_t^{\mathrm{aux}}|^2\,\mathrm{d}t + 4\,\langle \widetilde{Y}_t - Y_t^{\mathrm{aux}}, \nabla \log p_{T-t}(\widetilde{Y}_t) - \nabla \log p_{T-t}(Y_t^{\mathrm{aux}})\rangle\,\mathrm{d}t \\
&\quad + 4\,\langle \widetilde{Y}_t - Y_t^{\mathrm{aux}}, \nabla \log p_{T-t}(Y_t^{\mathrm{aux}}) - s(T-t, \hat{\theta}, Y_t^{\mathrm{aux}})\rangle\,\mathrm{d}t.
\end{aligned}
\tag{58}
$$

By integrating and taking the expectation on both sides in 58, using Corollary 14, Young's inequality with $\zeta \in (0, 1)$ and Assumption 4, we have

$$
\begin{aligned}
\mathbb{E}[|\widetilde{Y}_{T-\epsilon} - Y_{T-\epsilon}^{\mathrm{aux}}|^2] &= 2\int_0^{T-\epsilon} \mathbb{E}[|\widetilde{Y}_s - Y_s^{\mathrm{aux}}|^2]\,\mathrm{d}s \\
&\quad + 4\int_0^{T-\epsilon} \mathbb{E}[\langle \widetilde{Y}_s - Y_s^{\mathrm{aux}}, \nabla \log p_{T-s}(\widetilde{Y}_s) - \nabla \log p_{T-s}(Y_s^{\mathrm{aux}})\rangle]\,\mathrm{d}s \\
&\quad + 4\int_0^{T-\epsilon} \mathbb{E}[\langle \widetilde{Y}_s - Y_s^{\mathrm{aux}}, \nabla \log p_{T-s}(Y_s^{\mathrm{aux}}) - s(T-s, \hat{\theta}, Y_s^{\mathrm{aux}})\rangle]\,\mathrm{d}s \\
&\leq \int_0^{T-\epsilon} 2(1+\zeta)\,\mathbb{E}[|\widetilde{Y}_s - Y_s^{\mathrm{aux}}|^2]\,\mathrm{d}s \\
&\quad - 4\int_0^{t_J} \beta_{T-s}^{\mathrm{OS}}\mathbb{E}\left[|\widetilde{Y}_s - Y_s^{\mathrm{aux}}|^2\right]\,\mathrm{d}t + 2\zeta^{-1}\varepsilon_{\mathrm{SN}} \\
&\leq 2e^{2(1+\zeta)(T-\epsilon) - 4\int_0^{t_J} \beta_{T-t}^{\mathrm{OS}}\,\mathrm{d}t}\,\zeta^{-1}\varepsilon_{\mathrm{SN}}.
\end{aligned}
\tag{59}
$$

Using 59, Remark 15, and $t_J = T - \epsilon$, we have

$$
\begin{aligned}
W_2(\mathcal{L}(\widetilde{Y}_{t_J}), \mathcal{L}(Y_{t_J}^{\mathrm{aux}})) &\leq \sqrt{\mathbb{E}[|\widetilde{Y}_{t_J} - Y_{t_J}^{\mathrm{aux}}|^2]} \\
&\leq \sqrt{2\zeta^{-1}}e^{(1+\zeta)(T-\epsilon) - 2\int_\epsilon^T \beta_t^{\mathrm{OS}, K, \mu}\,\mathrm{d}t}\sqrt{\varepsilon_{\mathrm{SN}}}.
\end{aligned}
\tag{60}
$$

**Upper bound on** $W_2(\mathcal{L}(Y_{t_J}^{\mathbf{aux}}), \mathcal{L}(\widehat{Y}_t^{\mathbf{EM}}))$. Using Itô's formula, we have, for $t \in [0, T - \epsilon]$,

$$
\begin{aligned}
\mathrm{d}|Y_t^{\mathrm{aux}} &- \widehat{Y}_t^{\mathrm{EM}}|^2 \\
&= 2\langle Y_t^{\mathrm{aux}} - \widehat{Y}_t^{\mathrm{EM}}, Y_t^{\mathrm{aux}} + 2\,s(T-t, \hat{\theta}, Y_t^{\mathrm{aux}}) - \widehat{Y}_{\lfloor t/\gamma\rfloor\gamma}^{\mathrm{EM}} - 2\,s(T - \lfloor t/\gamma\rfloor\gamma, \hat{\theta}, \widehat{Y}_{\lfloor t/\gamma\rfloor\gamma}^{\mathrm{EM}})\rangle\,\mathrm{d}t \\
&= 2|Y_t^{\mathrm{aux}} - \widehat{Y}_t^{\mathrm{EM}}|^2\,\mathrm{d}t + 2\langle Y_t^{\mathrm{aux}} - \widehat{Y}_t^{\mathrm{EM}}, \widehat{Y}_t^{\mathrm{EM}} - \widehat{Y}_{\lfloor t/\gamma\rfloor\gamma}^{\mathrm{EM}}\rangle\,\mathrm{d}t \\
&\quad + 4\langle Y_t^{\mathrm{aux}} - \widehat{Y}_t^{\mathrm{EM}}, s(T-t, \hat{\theta}, Y_t^{\mathrm{aux}}) - s(T-t, \hat{\theta}, \widehat{Y}_t^{\mathrm{EM}})\rangle\,\mathrm{d}t \\
&\quad + 4\langle Y_t^{\mathrm{aux}} - \widehat{Y}_t^{\mathrm{EM}}, s(T-t, \hat{\theta}, \widehat{Y}_t^{\mathrm{EM}}) - s(T - \lfloor t/\gamma\rfloor\gamma, \hat{\theta}, \widehat{Y}_{\lfloor t/\gamma\rfloor\gamma}^{\mathrm{EM}})\rangle\,\mathrm{d}t.
\end{aligned}
\tag{61}
$$

Integrating and taking the expectation on both sides in 61, using Young's inequality for $\zeta \in (0,1)$, Cauchy Schwarz inequality, Assumption 3.a, Lemma 25, and Remark 1, we have

$$
\begin{aligned}
\mathbb{E}\left[|Y_{T-\epsilon}^{\mathrm{aux}} - \widehat{Y}_{T-\epsilon}^{\mathrm{EM}}|^2\right] &\leq (2+3\zeta)\int_0^{T-\epsilon}\mathbb{E}[|Y_t^{\mathrm{aux}} - \widehat{Y}_t^{\mathrm{EM}}|^2]\,\mathrm{d}t + \zeta^{-1}\int_0^{T-\epsilon}\mathbb{E}[|\widehat{Y}_t^{\mathrm{EM}} - \widehat{Y}_{\lfloor t/\gamma\rfloor\gamma}^{\mathrm{EM}}|^2]\,\mathrm{d}t \\
&\quad + 4\mathsf{K}_3(1+2T^\alpha)\int_0^{T-\epsilon}\mathbb{E}[|Y_t^{\mathrm{aux}} - \widehat{Y}_t^{\mathrm{EM}}|^2]\mathrm{d}t \\
&\quad + 2\zeta^{-1}\int_0^{T-\epsilon}\mathbb{E}[|s(T-t,\hat\theta,\widehat{Y}_t^{\mathrm{EM}}) - s(T-\lfloor t/\gamma\rfloor\gamma,\hat\theta,\widehat{Y}_{\lfloor t/\gamma\rfloor\gamma}^{\mathrm{EM}})|^2]\mathrm{d}t \\
&\leq (2+3\zeta+4\mathsf{K}_3(1+2T^\alpha))\int_0^{T-\epsilon}\mathbb{E}[|Y_t^{\mathrm{aux}} - \widehat{Y}_t^{\mathrm{EM}}|^2]\,\mathrm{d}t \\
&\quad + \zeta^{-1}\gamma(T-\epsilon)C_{\mathsf{EMose},2} + 8\zeta^{-1}\gamma^{2\alpha}(T-\epsilon)\mathsf{K}_1^2(1+4\mathbb{E}[|\hat\theta|^2]) \\
&\quad + 4\zeta^{-1}\mathsf{K}_3^2(1+2T^\alpha)^2\int_0^{T-\epsilon}\mathbb{E}[|\widehat{Y}_t^{\mathrm{EM}} - \widehat{Y}_{\lfloor t/\gamma\rfloor\gamma}^{\mathrm{EM}}|^2]\mathrm{d}t \\
&\leq (2+3\zeta+4\mathsf{K}_3(1+2T^\alpha))\int_0^{T-\epsilon}\mathbb{E}[|Y_t^{\mathrm{aux}} - \widehat{Y}_t^{\mathrm{EM}}|^2]\,\mathrm{d}t \\
&\quad + \zeta^{-1}\gamma(T-\epsilon)C_{\mathsf{EMose},2}(1+4\mathsf{K}_3^2(1+2T^\alpha)^2) \\
&\quad + 8\zeta^{-1}\gamma^{2\alpha}(T-\epsilon)\mathsf{K}_1^2(1+8\widetilde{\varepsilon}_{\mathrm{AL}}+8|\theta^*|^2) \\
&\leq e^{(2+3\zeta+4\mathsf{K}_3(1+2T^\alpha))(T-\epsilon)} \\
&\quad \times \Bigg(\zeta^{-1}\gamma(T-\epsilon)C_{\mathsf{EMose},2}(1+4\mathsf{K}_3^2(1+2T^\alpha)^2) \\
&\qquad\quad + 8\zeta^{-1}\gamma^{2\alpha}(T-\epsilon)\mathsf{K}_1^2(1+8\widetilde{\varepsilon}_{\mathrm{AL}}+8|\theta^*|^2)\Bigg).
\end{aligned}
\tag{62}
$$

Using 62 and $t_J = T - \epsilon$, we have

$$
\begin{aligned}
W_2(\mathcal{L}(Y_{T-\epsilon}^{\mathrm{aux}}), \mathcal{L}(\widehat{Y}_{T-\epsilon}^{\mathrm{EM}})) &\leq \gamma^{1/2}\zeta^{-1/2}(T-\epsilon)^{1/2}e^{(1+(3/2)\zeta+2\mathsf{K}_3(1+2T^\alpha))(T-\epsilon)} \\
&\quad \times (C_{\mathsf{EMose},2}^{1/2}(1+2\mathsf{K}_3(1+2T^\alpha)) + 2\sqrt{2}\mathsf{K}_1(1+8\widetilde{\varepsilon}_{\mathrm{AL}}+8|\theta^*|^2)^{1/2}).
\end{aligned}
\tag{63}
$$

**Final upper bound on $W_2(\mathcal{L}(Y_J^{\mathbf{EM}}), \pi_{\mathsf{D}})$.** Substituting 53, 57, 60, and 63 into 51, we have

$$
\begin{aligned}
W_2(\mathcal{L}(Y_J^{\mathrm{EM}}), \pi_{\mathsf{D}}) &\leq (\sqrt{\mathbb{E}[|X_0|^2]} + \sqrt{d})2\sqrt{\epsilon} \\
&\quad + \sqrt{2}(\sqrt{\mathbb{E}[|X_0|^2]} + \sqrt{d})e^{-2\int_\epsilon^T \beta_t^{\mathrm{OS},K,\mu}\,\mathrm{d}t-\epsilon} \\
&\quad + \sqrt{2\zeta^{-1}}e^{(1+\zeta)(T-\epsilon)-2\int_\epsilon^T \beta_t^{\mathrm{OS},K,\mu}\,\mathrm{d}t}\sqrt{\varepsilon_{\mathrm{SN}}} \\
&\quad + \gamma^{1/2}\zeta^{-1/2}(T-\epsilon)^{1/2}e^{(1+(3/2)\zeta+2\mathsf{K}_3(1+2T^\alpha))(T-\epsilon)} \\
&\quad \times (C_{\mathsf{EMose},2}^{1/2}(1+2\mathsf{K}_3(1+2T^\alpha)) + 2\sqrt{2}\mathsf{K}_1(1+8\widetilde{\varepsilon}_{\mathrm{AL}}+8|\theta^*|^2)^{1/2}).
\end{aligned}
\tag{64}
$$

The bound for $W_2(\mathcal{L}(\widehat{Y}_J^{\mathrm{EM}}), \pi_{\mathsf{D}})$ in 64 can be made arbitrarily small by appropriately choosing parameters including $\epsilon, T, \varepsilon_{\mathrm{SN}}$ and $\gamma$. More precisely, for any $\delta > 0$, we first choose $0 < \epsilon < \epsilon_\delta$ with $\epsilon_\delta$ given in Table 3 such that the first term on the right-hand side of 64 is

$$
(\sqrt{\mathbb{E}[|X_0|^2]} + \sqrt{d})2\sqrt{\epsilon} < \delta/4.
\tag{65}
$$

Next, we choose $T > T_\delta$ with $T_\delta$ given in Table 3 such that the second term on the right-hand side of 64 is

$$
\sqrt{2}(\sqrt{\mathbb{E}[|X_0|^2]} + \sqrt{d})e^{-2\int_\epsilon^T \beta_t^{\mathrm{OS},K,\mu}\,\mathrm{d}t-\epsilon} < \delta/4.
\tag{66}
$$

Next, we turn to the third term on the right-hand side of 64. We choose $0 < \varepsilon_{\mathrm{SN}} < \varepsilon_{\mathrm{SN},\delta}$ with $\varepsilon_{\mathrm{SN},\delta}$ given in Table 3 such that

$$\sqrt{2\zeta^{-1}} e^{(1+\zeta)(T-\epsilon)-2\int_\epsilon^T \beta_t^{\mathrm{OS},K,\mu}\,\mathrm{d}t} \sqrt{\varepsilon_{\mathrm{SN}}} < \delta/4. \tag{67}$$

Finally, we choose $0 < \gamma < \gamma_\delta$ with $\gamma_\delta$ given in Table 3 such that the fourth term on the right-hand side of 64 is

$$\begin{aligned}
\gamma^{1/2}\zeta^{-1/2}(T-\epsilon)^{1/2}&e^{(1+(3/2)\zeta+2\mathsf{K}_3(1+2T^\alpha))(T-\epsilon)}\\
&\times (C_{\mathsf{EMose},2}^{1/2}(1+2\mathsf{K}_3(1+2T^\alpha)) + 2\sqrt{2}\mathsf{K}_1(1+8\widetilde{\varepsilon}_{\mathrm{AL}}+8|\theta^*|^2)^{1/2}) < \delta/4.
\end{aligned} \tag{68}$$

Using 65, 66, 67, and 68, we obtain $W_2(\mathcal{L}(\widehat{Y}_J^{\mathrm{EM}}), \pi_{\mathsf{D}}) < \delta$. □

*Proof of Theorem 21.* Using the splitting 51, the proof follows along the same lines of the Proof of Theorem 19 for the estimation of the error bounds of the terms $W_2(\pi_{\mathsf{D}}, \mathcal{L}(Y_{t_J}))$, $W_2(\mathcal{L}(Y_{t_J}), \mathcal{L}(\widetilde{Y}_{t_J}))$, and $W_2(\mathcal{L}(\widetilde{Y}_{t_J}), \mathcal{L}(Y_{t_J}^{\mathrm{aux}}))$. The error bound for $W_2(\mathcal{L}(Y_{t_J}^{\mathrm{aux}}), \mathcal{L}(Y_J^{\mathrm{EM}}))$ is derived along the same lines of Bruno et al. (2025, Proof of Theorem 10). Putting these four estimates together leads to 27 and 28. □

## D  Modified Half-Normal Distribution

In this section, we recall the probability density function of the modified half-normal distribution, see e.g., Sun et al. (2023), used in Section 3.4.1 and defined as

$$g(x) = \frac{2\xi^{\frac{\upsilon}{2}} x^{\upsilon-1} \exp\left(-\xi x^2 + \psi x\right)}{\Psi\left(\frac{\upsilon}{2}, \frac{\psi}{\sqrt{\xi}}\right)}, \qquad x \geq 0, \tag{69}$$

where $\upsilon, \xi > 0$, $\psi \in \mathbb{R}$, and the normalizing constant

$$\Psi\left(\frac{\upsilon}{2}, \frac{\psi}{\sqrt{\beta}}\right) := \sum_{n=0}^\infty \frac{\Gamma\left(\frac{\upsilon}{2} + \frac{n}{2}\right)}{\Gamma(n)} \frac{\psi^n \xi^{-n/2}}{n!},$$

is the Fox–Wright function (Fox, 1928; Wright, 1935). We point out that the half-normal distribution, truncated normal distribution, gamma distribution, and square root of the gamma distribution are all special cases of the modified Half-Normal distribution 69. The distribution 29 follows by taking the symmetric extension of 69, i.e. $g(|x|)/2$, and choosing $\upsilon = 1$ and $\psi = -1$.

## E  Table of Constants

Table 3 displays full expressions for constants which appear in Theorem 19 and Theorem 21.

Table 3: Explicit expressions for the constants in Theorem 19 and Theorem 21.

| CONSTANT | DEPENDENCY | FULL EXPRESSION |
|---|---|---|
| $C_1$ | $O(\sqrt{d})$ | $2(\sqrt{\mathbb{E}[|X_0|^2]} + \sqrt{d})$ |
| $C_2$ | $O(\sqrt{d})$ | $\sqrt{2}\left(\sqrt{\mathbb{E}[|X_0|^2]} + \sqrt{d}\right)$ |
| $C_3(T,\epsilon)$ | $O(e^{(1+\zeta)(T-\epsilon)-2\int_\epsilon^T \beta_t^{\mathrm{OS},K,\mu}\,dt})$ | $\sqrt{2\zeta^{-1}}e^{(1+\zeta)(T-\epsilon)-2\int_\epsilon^T \beta_t^{\mathrm{OS},K,\mu}\,dt}$ |
| $C_{\mathrm{EM},2}(T)$ | $O(Me^{T^{2\alpha+1}}T^{2\alpha+1}\widetilde{\varepsilon}_{\mathrm{AL}})$ | $e^{T(4+8\mathsf{K}_{\mathrm{Total}}^2(1+T^{2\alpha}))}$ $\times (\mathbb{E}[|\widehat{Y}_0^{\mathrm{EM}}|^2] + 16\mathsf{K}_{\mathrm{Total}}^2 T(1+2\widetilde{\varepsilon}_{\mathrm{AL}}+2|\theta^*|^2)(1+T^{2\alpha})+2dT)$ |
| $C_{\mathrm{EM},4}(T)$ | $O(d^2 e^{T^{4\alpha+1}}T^{4\alpha+1})$ | $e^{T(\frac{21}{2}+128\mathsf{K}_{\mathrm{Total}}^4(1+T^{4\alpha}))}$ $\times (\mathbb{E}[|\widehat{Y}_0^{\mathrm{EM}}|^4] + 1024\mathsf{K}_{\mathrm{Total}}^4 T(1+\mathbb{E}[|\hat{\theta}|^4])(1+T^{4\alpha})+8(d^2+4d+4)T)$ |
| $C_{\mathrm{EMose},2}$ | $O(de^{T^{2\alpha+1}}T^{4\alpha+1}\widetilde{\varepsilon}_{\mathrm{AL}})$ | $2(C_{\mathrm{EM},2}(T) + \mathsf{K}_{\mathrm{Total}}^2(1+T^{2\alpha})(16C_{\mathrm{EM},2}(T)+32(1+2\widetilde{\varepsilon}_{\mathrm{AL}}+2|\theta^*|^2)))+2d$ |
| $C_4(T,\epsilon)$ | $O(\sqrt{d}e^{T^{2\alpha+1}}T^{3\alpha+1}\widetilde{\varepsilon}_{\mathrm{AL}}^{1/2})$ | $\zeta^{-1/2}(T-\epsilon)^{1/2}e^{(1+(3/2)\zeta+2\mathsf{K}_3(1+2T^\alpha))(T-\epsilon)}$ $\times (C_{\mathrm{EMose},2}^{1/2}(1+2\mathsf{K}_3(1+2T^\alpha))+2\sqrt{2}\mathsf{K}_1(1+8\widetilde{\varepsilon}_{\mathrm{AL}}+8|\theta^*|^2)^{1/2})$ |
| $C_{\mathrm{EMose},4}$ | $O(d^2 e^{T^{4\alpha+1}}T^{8\alpha+1})$ | $8(C_{\mathrm{EM},4}(T) + \mathsf{K}_{\mathrm{Total}}^4(1+T^{4\alpha})(1024C_{\mathrm{EM},4}(T)+8192(1+\mathbb{E}[|\hat{\theta}|^4])))+144d^2$ |
| $\widetilde{C}_4(T,\epsilon)$ | $O(de^{T^{4\alpha+1}}T^{4\alpha+1}\widetilde{\varepsilon}_{\mathrm{AL}}^{1/4})$ | $\sqrt{2}e^{2(1+\zeta+\mathsf{K}_3(1+2T^\alpha+4\mathsf{K}_3(1+4T^{2\alpha})))(T-\epsilon)}\sqrt{T-\epsilon}$ $\times\left(\mathsf{K}_4^2\zeta^{-1}(1+4T^{2\alpha})C_{\mathrm{EMose},4}+4d(1+8\mathsf{K}_3^2(1+4T^{2\alpha}))\right.$ $+2\zeta^{-1}\mathsf{K}_1^2(1+8(\widetilde{\varepsilon}_{\mathrm{AL}}+|\theta^*|^2))$ $+4\zeta^{-1}d(1+8\mathsf{K}_3^2(1+4T^{2\alpha}))$ $\times[(1+16\mathsf{K}_{\mathrm{Total}}^2(1+T^{2\alpha}))C_{\mathrm{EM},2}(T)$ $+32\mathsf{K}_{\mathrm{Total}}^2(1+T^{2\alpha})(1+2\widetilde{\varepsilon}_{\mathrm{AL}}+2|\theta^*|^2)]$ $+2[(1+8\mathsf{K}_3^2(1+4T^{2\alpha}))^{1/2}C_{\mathrm{EMose},2}^{1/2}+2\mathsf{K}_1(1+8\widetilde{\varepsilon}_{\mathrm{AL}}+8|\theta^*|^2)^{1/2}]$ $\left.\times[d\sqrt{2}(1+8\mathsf{K}_3^2(1+4T^{2\alpha}))^{1/2}]\right)^{1/2}$ |
| $\epsilon_\delta$ | - | $\delta^2/(64(\sqrt{\mathbb{E}[|X_0|^2]}+\sqrt{d})^2)$ |
| $T_\delta$ | - | Obtained solving $T > T_\delta$ using Proposition 16, i.e., $\ln(\mu(e^{2T}-1)+1)+(K/\mu+1)/(\mu(e^{2T}-1)+1)$ $> \ln(4\sqrt{2}((\mathbb{E}[|X_0|^2])^{1/2}+\sqrt{d})/\delta)+2\int_0^\epsilon \beta_t^{\mathrm{OS},K,\mu}\,dt + K/\mu+1-\epsilon$ |
| $\varepsilon_{\mathrm{SN},\delta}$ | - | $(\delta^2\zeta/32)e^{-2(1+\zeta)(T-\epsilon)+4\int_\epsilon^T \beta_t^{\mathrm{OS},K,\mu}\,dt}$ |
| $\gamma_\delta$ | - | $(\delta^2\zeta/16)(T-\epsilon)^{-1}e^{-2(1+(3/2)\zeta+2\mathsf{K}_3(1+2T^\alpha))(T-\epsilon)}$ $\times (C_{\mathrm{EMose},2}^{1/2}(1+2\mathsf{K}_3(1+2T^\alpha))+2\sqrt{2}\mathsf{K}_1(1+8\widetilde{\varepsilon}_{\mathrm{AL}}+8|\theta^*|^2)^{1/2})^{-2}$ |
| $\widetilde{\gamma}_\delta$ | - | $\min\left\{(\delta/(4\sqrt{2}))^{1/\alpha}(T-\epsilon)^{-1/(2\alpha)}e^{-(2/\alpha)(1+\zeta+\mathsf{K}_3(1+2T^\alpha+4\mathsf{K}_3(1+4T^{2\alpha})))(T-\epsilon)}\right.$ $\times\left(\mathsf{K}_4^2\zeta^{-1}(1+4T^{2\alpha})C_{\mathrm{EMose},4}+4d(1+8\mathsf{K}_3^2(1+4T^{2\alpha}))\right.$ $+2\zeta^{-1}\mathsf{K}_1^2(1+8(\widetilde{\varepsilon}_{\mathrm{AL}}+|\theta^*|^2))$ $+4\zeta^{-1}d(1+8\mathsf{K}_3^2(1+4T^{2\alpha}))$ $\times[(1+16\mathsf{K}_{\mathrm{Total}}^2(1+T^{2\alpha}))C_{\mathrm{EM},2}(T)$ $+32\mathsf{K}_{\mathrm{Total}}^2(1+T^{2\alpha})(1+2\widetilde{\varepsilon}_{\mathrm{AL}}+2|\theta^*|^2)]$ $+2[(1+8\mathsf{K}_3^2(1+4T^{2\alpha}))^{1/2}C_{\mathrm{EMose},2}^{1/2}+2\mathsf{K}_1(1+8\widetilde{\varepsilon}_{\mathrm{AL}}+8|\theta^*|^2)^{1/2}]$ $\left.\left.\times[d\sqrt{2}(1+8\mathsf{K}_3^2(1+4T^{2\alpha}))^{1/2}]\right)^{-1/(2\alpha)},1\right\}$ |

Table 4 provides the main notation used throughout this work and indicates where each symbol is introduced.

Table 4: List of the main notation.

| SYMBOL | REFERENCE IN THE TEXT |
|---|---|
| $\pi_{\mathrm{D}}$ | Equation 1 |
| $\hat{\theta}$ | Assumption 1 |
| $\widetilde{\varepsilon}_{\mathrm{AL}}$ | Assumption 1 |
| $\partial U$ | Definition 3 |
| $K$ | Assumption 2-(ii) |
| $\mu$ | Assumption 2-(iii) |
| $\alpha$ | Assumption 3.a |
| $\varepsilon_{\mathrm{SN}}$ | Assumption 4 |
| $\kappa_U$ | Definition 11 |
| $f_L$ | Equation 13 |
| $\beta_t^{\mathrm{OS},K,\mu}$ | Equation 22 |
| $B(t,0,\mu,K)$ | Equation 23 |

