# OpenReview forum: "Wasserstein Convergence of Score-based Generative Models under Semiconvexity and Discontinuous Gradients"
_TMLR — Accepted by TMLR_

### Review · Reviewer_Zoog · 2025-06-22

**Summary Of Contributions:**

This paper establishes bounds on the Wasserstein-2 distance in Score-based Generative Models—specifically, between the data distribution and the final step of the backward process -- under the mild assumption that the negative log of the data distribution is weakly convex (and possibly non-differentiable). The bound presented in Theorem 13 is optimal with respect to the data dimension, scaling as $O(\sqrt{d})$. Additionally, Theorem 15 provides an improved bound in terms of the discretization step size $\gamma$, assuming Lipschitz smoothness of the score approximation.

**Audience:**

Yes

**Claims And Evidence:**

Yes

**Requested Changes:**

- Page 2: The sentence "...our estimates in Theorems 13 and Theorem 15 are derived without imposing any restrictions
on the stepsize of the generative algorithm" deserves some comments on which restrictions on the stepsizes in the previous works are. Maybe you can also include this information in Table 1.

- Page 3: About the mild assumptions on the target distribution that admits the backward process (3): Please specify what the assumptions are and how they effect your main assumptions. Because this paper is all about "mild assumptions", I do not think we can ignore even this assumption.

- Page 5: about the selector of the Frechet subdifferential, $h(x) \in \partial U(x)$: It seems we also need $h$ to be regular enough, i.e., Borel-measurable. Under which conditions such a measurable selector exists?

- Assumption 2: It seems to me that the R's in Assumption 2(ii) and (iii) need to agree, i.e., same R. If yes, please add this extra detail to the assumption.

- Remark 2: $\nabla p_t(x)$ is continuous for $t$ and $x$. Do you mean it is continuous w.r.t. x for fixed t, or it is continuous w.r.t. both variables?

- Assumption 3.a.: It is worth discussing the condition that $\alpha \in [1/2,1]$. I noticed you used the same condition as in [Bruno et al. 2025] but it is more complete if there are some comments in this paper.

- Definition 6 and Proposition 8: appropriate citations are needed to show the equivalence between two definitions of weak convexity. Also, in Proposition 8, (i) can you give a comment on why $\tilde{\mu} > 0$ and (ii) make sure the $R$ in item 2 is the same as the R in item 1 -- it is not obvious to me, especially for the case $\mu > L$.

- Corollary 9: I think we have to exclude $0$ in $[0,T]$ since the gradient of $\log p_0$ does not exist.

- Remark 10: I have difficulties understanding this one. (1) The statement "This gap may vanish for different functions..." needs citations; (2) If the gap can vanish, why don't you pick that function from the beginning? Because when you replace $L$ with $K+\mu$, the inequality (19) no longer holds (the inequality sign is flipped). Please provide more explanations in this regard and discuss how your results are still valid.

- About the critical time $t^*$ that the distribution evolves from non-log-concave to log-concave, I think a numerical illustration is very helpful for the readers' understanding.

- Related works: I found the following work very relevant to your work: Wang, Xixian, and Zhongjian Wang. "Wasserstein Bounds for generative diffusion models with Gaussian tail targets." arXiv preprint arXiv:2412.11251 (2024).

# Minor:

- Proposition 8: You used $L$ without giving its connection to Assumption 2, i.e., in Assumption 2, there are only $\mu, K, R$.

- Theorems 13 and 15: you are using $K$ as an index, viz., $Y_K^{EM}$. Note that $K$ is already used as the weak convexity modulus throughout the paper. I suggest using another character.

**Strengths And Weaknesses:**

# Strengths:

- The contribution is important in understanding score-based generative models in the context of nonsmooth, non-log-concave data distributions. Previous works have to assume either log of the data distribution is concave or twice continuously differentiable.

- The paper is generally well-written;  The motivation discussion and the contributions are pretty clear -- a detailed comparison with previous works is presented in Table 1.

# Weaknesses:

- The paper focuses solely on the Ornstein-Uhlenbeck process. I wondered what the difficulties would be when extending to other probability paths, or, at least, to more general SDEs.

- Assumption 3.b. (needed for Theorem 15) looks strong to me: the bound needs to hold uniformly for $\theta$ which is the neural network's parameters. I do not think it would be the case in practice. I wondered if one can relax this assumption by also allowing some growth for $\theta$ in the bound.

- Some mathematical details need a more careful revision; some recent references are missing -- See Requested Changes.

---

> ### Author Response · Authors · 2025-07-20
>
> Thank you for taking the time to review our submission and for your constructive feedback.
>
> Requested Changes
> 1. We have added several clarifications regarding the stepsize restrictions in prior works using the $W_2$ metric: (a) A footnote in the Introduction (page 2); (b) An expanded explanation in the Related Work and Comparison section (page 13); (c) A new comparison table (Table 2) summarizing the stepsize conditions employed in prior works alongside those used in our analysis.
> 2. We revised the text on page 3 to clarify the assumptions required on the target data distribution for the diffusion process to be well-defined under time reversal, specifically for the Ornstein–Uhlenbeck process.
> 3. This follows from the semiconvexity condition (Assumption 2 (ii)), as each element of the subdifferential set satisfies a one-sided Lipschitz property (see Lemma 4  and Corollary 5). This property ensures the existence of a selector. We have now clarified this point on page 5 before stating Assumption 2.
> 4. In Assumption 2, the constant $R$ appearing in parts  (ii) and (iii) is the same, and is therefore denoted using the same letter $R$. This has been clarified in the updated version.
> 5. We revised Remark 6 to explicitly state that the map $ (t,x) \mapsto p_t(x)$ defined over $(0,T] \times \mathbb{R}^d$,  belongs to $C^{1,2}((0,T] \times \mathbb{R}^d)$.
> 6. We have expanded Remark 7 to discuss the role of the Hölder index $\alpha$. We cited Assumption 3.a from [Bruno et al., TMLR 2025] on page 6, before introducing our own, to complement the existing citation in Remark 7.
> 7. We added comments on page 7 referencing Definition 10 (adapted from the weak convexity notion used in ergodicity studies of gradient flow SDEs) and Proposition 13 (derived in Appendix B, pages 20-21). The case $\tilde{\mu} > 0$ is explicitly addressed in the proof of Proposition 13 in Appendix B (page 21). A corresponding footnote in the text of the proposition has been added. For point (ii), we clarified that the constant $R$ in item 2 is the same as in item 1. This is further substantiated in the proof of Proposition 13 in Appendix B (pages 20-21), which shows the relationship between these items, including the case $\mu > L$.
> 8. We have fixed the typo in Corollary 14, and used $t \in (0,T]$.
> 9. In the updated Remark 15, we added references for point (1) regarding the function class $\widetilde{\mathcal{F}}$. For point (2):  We have not investigated which particular $f \in \widetilde{\mathcal{F}}$ can be chosen to make $\beta_0^{0S} =  -K$.  However, the limit in equation (21) in Remark 15 shows that $ - \beta_0^{0S}$ is not the lowest monotonicity bound for the right-hand side of equation (19) in Corollary 14. To obtain the monotonicity bound under our Assumption 2, we used the version from Lemma 5.9 in [Conforti et al, 2023b], which is adapted in Lemma 12. This result employes the specific function  $f_L(r)= 2 L^{1/2} \tanh((rL^{1/2})/2)$, which is commonly used in coupling-based analyses of Fokker--Planck equations. Proposition 13 links  Assumption 2 to the definition weak convexity involving $f_L(r)$, which in turn supports the derivation of the bound $\beta_t^{\text{OS}}$ in Corollary 14. We clarified this connection in the updated version.
> 10. We now illustrate the time behaviour of the score function for a Gaussian mixture  with two equally weighted components having the same variance. We computed the time $\bar{t} <t^{\star}$ in the new Remark 18.
> 11. We have included the recommended work in both the "Introduction" (page 2) and the "Related Work and Comparison" section (page 12).
>
> Minor
> 1. Proposition 13 links Assumptions 2(ii) and 2(iii) and the notion of weak convexity, as formalized in Definition 10 via the constant $L$. We have added an explicit reference to this connection in the text of Proposition 13.
> 2. We have replaced the index $K$ with $J$, so that $Y_J^{\text{EM}}$ no longer conflicts with the weak convexity constant $K$.
>
> Weaknesses
> 1. We have chosen to focus on the Ornstein-Uhlenbeck process to make the main results easier to interpret. Extending the analysis to more general SDEs would have required tracking additional dependencies in the error bounds arising from the drift and diffusion coefficients of the forward process. Analogous results in the  $W_2$-convergence literature also adopt the Ornstein-Uhlenbeck as the forward process, e.g. [Gentiloni-Silveri and Ocello, ICML 2025], [Bruno et al, 2025, TMLR], [Yu and Yu, 2025], and [Wang and Wang, 2024].
> 2. Comparable results rely on similar or stronger assumptions: Assumption H5 in [Strasman et al, TMLR 2025], Assumption 4 in [Wang and Wang, 2024], Remark 3.4 in [Gentiloni-Silveri and Occello, ICML 2025], Assumption 3.b in [Bruno et al, 2025, TMLR]. Relaxing the condition on $\nabla_x s$ to allow growth in $\theta$ would require introducing additional assumptions on the optimizer $\hat{\theta}$.

---

### Review · Reviewer_qYo2 · 2025-07-02

**Summary Of Contributions:**

This paper proves non-asymptotic convergence guarantees in Wasserstein-2 distance for score-based generative models (SGMs) under relaxed assumptions. Specifically, the authors assume the data distribution has a semiconvex potential and allow for discontinuous gradients, removing the need for smoothness or strong log-concavity. They derive explicit bounds with optimal dimension dependence (`O(√d)`) and convergence rates (`O(γ^α)`), covering practical examples like Gaussian mixtures, double-well potentials, and elastic net.

**Audience:**

Yes

**Broader Impact Concerns:**

There are no major concerns. The work is theoretical, with no direct applications that would raise ethical or safety risks.

**Claims And Evidence:**

Yes

**Requested Changes:**

* Briefly explain the core assumptions (e.g., what semiconvexity and strong convexity at infinity mean) in more intuitive terms.
* Include a toy numerical example that shows convergence under non-smooth or multimodal targets.
* Consider adding a table of notation and constants, as there are many parameters across theorems.

**Strengths And Weaknesses:**

**Strengths**:  The strongest part of the paper is the theoretical contribution. It broadens the scope of SGM convergence guarantees to more realistic data distributions, which are often non-smooth in practice. The proofs are rigorous, and the results are cleanly stated with explicit constants.

**Weaknesses**
- The weaknesses are mostly around accessibility and empirical support. Some technical constructions (like the weak convexity profile and function `f_L`) could use more intuition or explanation. Readers not deeply familiar with convex analysis may find some sections hard to parse.
- Also, while this is a theory paper, a minimal empirical illustration (e.g., on synthetic 1D or 2D distributions) would help ground the results.

- Another potential issue is that while Assumptions 2 and 3 are weaker than those in prior work, they still require conditions that may not be easy to verify when using neural networks as score approximators.

---

> ### Author Response · Authors · 2025-07-20
>
> Thank you for taking the time to review our submission and for your constructive feedback. We answer your points made in the Section "Requested Changes" and "Strengths and weaknesses" below.
>
> "Requested Changes":
> 1. We have added a more intuitive explanation of the core assumptions and related concepts. Specifically: (a) page 5 in Section 3.1 for the notions of subdifferential and semiconvexity; (b)  A footnote in Assumption 2 (page 6) explains the concept of  strong convexity at infinity; (c)  Remark 11 (page 7) provides further clarification on the definition of weak convexity.
> 2. We have included a toy numerical example in Remark 18 (page 9), illustrating the time evolution of the score function for a Gaussian mixture with two equally weighted modes of equal variance. The example shows convergence toward the score function of the invariant measure and includes a computation of the time  $\bar{t}<t^{\star}$, as defined in Proposition 16 (page 9).
> 3. Table 3 (Appendix E,  page 27) provides the explicit expressions for all constants used in Theorems 19 and Theorem 21. We have included the new Table 4 (Appendix E, page 28), which summarizes the main notation used throughout the manuscript.
>
> "Weaknesses":
> 1. Please refer to points 1 and 2 in the "Requested Changes" section above.
> 2. Assumption 2 is easier to verify than the weak convexity condition in Assumption H1 of [Gentiloni-Silveri and Ocello,  ICML 2025], which requires computing the constant $L$ in $f_L$ that appears in the lower bound of the weak convexity profile. For example, in Section 3.4 (pages 10–11), we explicitly compute the semiconvexity constant $K$ and strong convexity at infinity constant $\mu$ for data distributions satisfying Assumption 2. We have also expanded Remark 7 (page 6), which discusses Assumption 3.a and incorporates the relevant points from Remark 6 in [Bruno et al, TMLR, 2025].

---

### Review · Reviewer_o7dK · 2025-07-07

**Summary Of Contributions:**

This work provides Wasserstein-2 convergence guarantees for score-based generative models (SGMs) targeting semiconvex data distributions with potentially non-smooth potentials. The theoretical analysis shows that it can achieve $\sqrt{d}$ dimension dependence rate in the stepsize term. It generalizes previous analysis to more complex non-smooth settings.

**Audience:**

Yes

**Claims And Evidence:**

Yes

**Requested Changes:**

Please add some introductory paragraphs before each theorem and lemma. For example, clearly explain the purpose and significance of Lemma 7. For Proposition 8, instead of jumping directly into the content, first motivate the result: explain why this proposition is needed, what problem it addresses, and how it contributes to the main theory.

**Strengths And Weaknesses:**

Strengths:

The paper is a theoretical paper, which provides detailed proof to support its claim. The studied problem: Wasserstein convergence for SGMs, is interesting and important.

Weaknesses:

1. There is some notation abuse. For example, in equation (11) $\partial U(x)$ is defined to be a set in equation (1). However, it is used as a vector and takes the inner product with another vector. It should be $\nabla U$ given that $U$ is differentiable or something else. Moreover, I find that Definition 6 directly relates to [1] without good reference or comparison (if they are different).

2. In Remark 10, I do not quite understand why you mention the specific function class, and why “the gap may vanish”? What do you mean?

3. The presentation of the main results (Theorems 13,15) is very ambiguous. It follows the writing of [2], replacing $L_{MO}$ with an integral of $\beta$. The presentation and remarks on those results puzzled me. First, the choice of $T$ arises from solving a complex equation. At first glance, it appears that $T$ could depend on $\log d$. Since many of the “constants” in the bounds involve terms like $e^{T^{2\alpha +1}}$, this could introduce additional dependence on $d$, casting doubt on the claim of “optimality” on the dimension. Second, it's unclear what notion of “optimal” is intended here. In prior work, optimality typically refers to the dependence on $d$ in terms of sample complexity or iteration complexity. However, your discussion only focuses on the rate term preceding the step $\gamma$. Moreover, optimality usually corresponds to a matching lower bound. Third, “resulting from numerical techniques introduced in Kumar & Sabanis (2019) and employed in the proof of Theorem 15 to achieve the optimal convergence rate of order $\alpha \in [1/2, 1]$”, appears to be directly copy-pasted from [2], and it's unclear what is actually being conveyed here.

4. As I have pointed out in 3, this paper highly resembles [2], except for a slightly weaker concave assumption. I don’t quite understand why this improvement is essential for this setting.

5. The paper makes several assumptions on the estimated score function. While similar assumptions have appeared in previous works, I believe some of them are too strong and less suitable compared to what is commonly adopted in the literature. In Assumption 3.1, it is unclear why the "Lipschitz constant" for the $\theta$ and $x$ terms is assumed to depend only on $t$, and not on 𝜃$θ$ or 𝑥$x$ themselves. This seems overly restrictive. Moreover, the parameter $\alpha \in [1/2, 1]$ appears somewhat artificial. In Assumption 4, the expectation is taken over an auxiliary process. However, this seems less appropriate than taking the expectation over the forward process, which is directly related to the training procedure. Since all training data and objectives are derived from the forward process, using the auxiliary process introduces a mismatch.

---

[1]  Gentiloni-Silveri & Ocello 2025, Beyond Log-Concavity and Score Regularity: Improved Convergence Bounds for Score-Based Generative Models in W2-distance

[2] Bruno et al., 2025, On diffusion-based generative models and their error bounds: The log-concave case with full convergence estimates

---

> ### Author Response · Authors · 2025-07-20
>
> Thank you for taking the time to review our submission and for your feedback.
>
> 1. $\nabla U$ may not exist everywhere. For this reason, we use the Fréchet subdifferential $\partial U(x)$. We have provided additional details in the updated version of Section 3.1 page 5. The Fréchet subgradient at $x$ is denoted by $h(x) \in \partial U(x)$. The notion of weak convexity appears in prior work on ergodicity of Langevin dynamics and noise-perturbed gradient flows, where it is applied to $\nabla U$. In contrast, we apply this concept to $h(x)$. This distinction is now made explicit in Definition 10, with added citations at the beginning Section 3.2.
> 2. We have expanded Remark 15 to clarify this point. The limit in equation (21) shows that the monotonicity bound $-\beta_t^{OS}$ is the not tightest possible lower bound because  $\beta_t^{OS}$ is derived via Lemma 12 and Proposition 13 using the definition of weak convexity (Definition 10) with the specific choice of $f_L(r)$ in eq. (13). The definition of weak convexity holds for functions in the broader class $\mathcal{\tilde{F}}$, which can yield $\beta_0^{0S}=-K$.
> 3. (Point 4). Section 3.1 (page 5) has been revised to better motivate semiconvexity. Semiconvex functions include non-convex, non-differentiable functions with bounded curvature from below-- a broader class than classical convex or strongly log-concave functions. This generalization is not merely theoretical: as emphasized through citations (page 5) and examples (Section 3.4), such functions naturally arises in optimization and ML, including in robust regression, non-smooth regularization, multimodal density estimation, and other problems where potentials may have discontinuous gradients or lack global smoothness. In contrast to existing results ([Bruno et al, TMLR, 2025], [Gao et al, JMLR 2025], [Strasman et al, TMLR 2025]), which assume (at least) strong log-concavity (implying both smoothness and convexity), semiconvexity allows for discontinuities and weaker curvature control, requiring generalized subgradients and new analytical techniques. Our work is the first to establish non-asymptotic $W_2$-convergence guarantees for SGMs under such weak regularity assumptions. This fills a significant gap in the literature and makes progress toward understanding SGMs in more realistic settings.
> 3. (Point 3) Under the weak assumptions on $\pi_{D}$ imposed by Assumption 2, a central challenge lies in controlling the error introduced when replacing the initial condition of the backward process  $Y_t$ eq. (2) with the invariant measure of the forward process in $\tilde{Y_t} $ eq. (3). This means establishing new contraction estimates in the $W_2$ metric between backward SDEs with different initializations, i.e.,  $Y_t$ and $\tilde{Y_t} $, since $L_{MO}$ from [Bruno et al, TMLR 2025] may no longer be strictly positive under our weaker conditions. This requires demonstrating a form of contraction for the score-based drift terms in both SDEs ensuring that the factor $e^{-2 \int_{\epsilon}^{T}\beta^{OS, K, \mu}_t dt -\epsilon}$ in eq. (26) remains sufficiently small. Ensuring this under such weak assumptions on the data distribution is mathematically non-trivial and constitutes a key technical contribution of our analysis. In the worst case, the time $t^*$ in Proposition 16 can be computed using standard numerical (root-finding) methods, based on the explicit expression provided in the proposition. Moreover, $T$ does not depend on $\log d$. Our work yields the best known upper bounds in $W_2$ in terms of the data dimension ($O(\sqrt{d})$) and convergence rate of order $\alpha \in [1/2,1]$. The notion of "optimal" is consistent with prior works, e.g., [Gentiloni-Silveri Ocello, ICML 2025], [Bruno et al, TMLR 2025]. Finally, we have updated Remark 22 to highlight the role of the explicit Milstein numerical scheme in achieving this convergence rate.
> 5. We have expanded Remark 7 to clarify the role of the Hölder index $\alpha$. We have included an explanation -along with a citation- on how Assumption 3.a is interpreted in the context of neural network-based approximations. Assumption 4 involves taking expectations with respect to an auxiliary process whose density is known, since the approximating function $s$, the estimator $\hat{\theta}$, and its initial condition are all known. Thus, the auxiliary process is easy to simulate. By contrast, expectation with respect to the forward process depends on the unknown data distribution. The discrepancy introduced by using the auxiliary process can be addressed, as explained in Remark 8 of [Bruno et al, TMLR 2025]. Assumption 4 has been discussed in the $W_2$ convergence literature as both practically verifiable and theoretically grounded. For completeness, we refer to Remark 3.4 in [Gentiloni-Silveri Ocello ICML, 2025], Remarks 7-8 in [Bruno et al, TMLR 2025], Assumption 3 in [Gao et al, JMLR 2025].
>
> Requested changes: This point has been addressed in the updated version.

---

### Author Response · Authors · 2025-07-20

We thank the reviewers for their insightful comments. We have uploaded a revised version of the paper with changes highlighted in red.

---

### Decision · Action_Editor_kngS · 2025-08-25

**Recommendation:** Accept as is

**Additional Comments:**

The authors provide an upper bound for the Wassestein distance scaling with the square root of the data dimension which is the best rate known even under stronger assumptions. In addition, the authors propose several settings used in practice and in other works where the main assumptions can be verified (semi-convexity on a ball.

 Reviewers found that the paper provides an interesting theoretical contribution to an important problem: analyzing score-based generative models under realistic assumptions. They were concerned by the motivations of the proposed assumptions and their links to previous works. They also asked for some clarifications on the theoretical results (dependency on the dimension, assumptions to be verified on the estimated score function). During rebuttal, the authors proposed important revisions to discuss their results with additional details and to position their work and assumptions with respect to other works. The additional remarks and extended discussions allow to better motivate the contribution which provides a generalization of existing results for score-based generative models important to a large audience in the ML community. The proposed semi-convex framework allows to consider non-convex cases and the authors highlight several settings  (non-smooth regularization, multimodal density estimation) of prractical interest.

**Audience:**

Yes

**Audience Explanation:**

Extending known results on score-based generative models is an important problem of interest to a large audience in the ML community.
Providing realistic assumptions and explicit upper bounds is a very active area of research perfectly adapted to TMLR's audience.

**Claims And Evidence:**

Yes

**Claims Explanation:**

This works provides a non-asymptotic convergence analysis of score-based generative models in Wasserstein distance. They authors aim at extending recent works where convergence rates were obtained under strong assumptions such as smoothness and strict log-concavity of the data distribution, or a twice continuously differentiable target potential and weakly convex data distribution.The authors provide non asymptotic convergence rates under  semiconvexity assumptions on the data distribution and without restrictive assumptions on the step size of the backward sampler as it is usually the case in other works.
Their claims are supported by strong theoretical results and detailed remarks on the applicability of the results.